# MODEL SOUPS NEED ONLY ONE INGREDIENT

## ABSTRACT

Fine-tuning large pre-trained models on a target distribution often improves in-distribution (ID) accuracy, but at the cost of out-of-distribution (OOD) robustness as representations specialize to the fine-tuning data. Weight-space ensembling methods, such as Model Soups, mitigate this effect by averaging multiple checkpoints, but they are computationally prohibitive, requiring the training and storage of dozens of fine-tuned models. In this paper, we introduce MonoSoup, a simple and data-free approach that achieves a strong ID–OOD balance using *only a single* checkpoint. Our method applies Singular Value Decomposition (SVD) to each layer's update, splitting it into high-energy directions that capture task-specific adaptation and low-energy directions that introduce noise but may still encode residual signals useful for robustness. MonoSoup then re-weights these components with adaptive, layer-wise coefficients that account for the spectral and geometric structure of the model. Experiments on CLIP models fine-tuned on ImageNet and evaluated under natural distribution shifts, as well as on Qwen language models tested on mathematical reasoning and multiple-choice benchmarks, show that this plug-and-play approach is a practical and effective alternative to multi-checkpoint methods, retaining much of their benefits without their computational overhead.

## 1 INTRODUCTION

The *pre-train-then-finetune* paradigm (Kumar et al., 2022) has become the de-facto approach for leveraging the capabilities of foundation models (Bommasani et al., 2022) and has accelerated progress across a wide range of applications (Radford et al., 2021; Rombach et al., 2022). However, specialization often comes at a cost: the fine-tuning process that adapts a model to a target distribution frequently degrades its general-purpose knowledge, leading to a significant drop in out-of-distribution (OOD) performance, a phenomenon known as *representation collapse* (Aghajanyan et al., 2020). This leads to a trade-off between in-distribution (ID) performance and OOD robustness, which remains a central challenge for the reliable deployment of these powerful models (Kumar et al., 2022; Goyal et al., 2023).

To address this trade-off, post-hoc methods that directly manipulate model weights have gained traction. A prominent example is Model Soups (Wortsman et al., 2022a), which improves both ID and OOD performance by averaging the weights of multiple fine-tuned models. While effective, this approach is often impractical due to the computational and storage overhead of training and retaining dozens of checkpoints. To reduce this burden, ModelStock (Jang et al., 2024) was proposed as a more efficient alternative, requiring only two models and weighting their updates according to geometric alignment. However, this assumption of having access to two suitable checkpoints is often unrealistic in practice, as model repositories typically store only a single, best-performing version. Furthermore, Wise-FT (Wortsman et al., 2022b) explored single-model robustness by interpolating between the pre-trained and fine-tuned weights, leveraging the low-loss path between them. While effective in tracing the trade-off between specialization and robustness, Wise-FT applies a uniform interpolation across all layers and directions, leaving finer-grained anisotropic effects unaddressed. These observations naturally raise the following question:

*"Can we retain the benefits of model soups when only a single fine-tuned model is available?"*

In this paper, we answer this question affirmatively. We begin by analyzing when common multi-model merging techniques succeed. Our analysis reveals that improvements in both ID

and OOD accuracy are consistently obtained when the fine-tuning updates of two models are well-aligned (high cosine similarity), highlighting a strong link between geometric similarity and generalization. To validate this insight, we introduce a Similarity-Filtered Greedy Soup that selects only those models that would improve the overall alignment of the soup. This simple variant both confirms our hypothesis and provides a data-free, computationally efficient alternative to standard soups. These findings align with a unifying principle from recent work (Wortsman et al., 2022a; Jang et al., 2024; Rame et al., 2023; Gargiulo et al., 2025; Stoica et al., 2024) that successful weight-space ensembling methods reinforce dominant directions that encode task-relevant signals, while suppressing noisy or misaligned directions that degrade both ID and OOD performance.

Building on these insights, we shift from analyzing pairs of models to studying the properties of a single fine-tuned checkpoint. Since such models often over-specialize at the cost of OOD robustness, our goal is to test whether the structural signals that enable successful merging across multiple models can also be exploited within a single model. To this end, we propose MonoSoup, which applies Singular Value Decomposition (SVD) to each layer's update and decomposes it into two complementary components: a principal subspace capturing high-energy directions associated with task-specific knowledge, and an orthogonal complement capturing low-energy directions that introduce noise but may still encode residual signals useful for robustness. MonoSoup re-weights these two components with principled, layer-wise coefficients that adapt to the structure of the model, yielding a single checkpoint that better balances specialization and generalization. Extensive experiments show that MonoSoup matches or exceeds the performance of multi-model methods while using just one fine-tuned checkpoint. On CLIP (Radford et al., 2021) models fine-tuned on ImageNet (Deng et al., 2009), it improves the average OOD accuracy of the strongest baseline by $\sim 1\%$ and recovers up to 8% on weaker, representation-collapsed checkpoints, while maintaining strong ID accuracy. Similar gains are also observed on language-based mathematical reasoning and QA tasks using Qwen (Yang et al., 2025). Moreover, MonoSoup complements single-model techniques such as Wise-FT (Wortsman et al., 2022b), providing a stronger checkpoint that further improves the ID–OOD trade-off when the two are combined.

Our contributions are the following:

1. We establish a geometric perspective on when model merging succeeds or fails, showing that performance is closely tied to the alignment of fine-tuning updates. This analysis clarifies principles underlying multi-model methods and motivates their extension to the single-checkpoint setting.

2. Based on this analysis, we introduce Similarity-Filtered Greedy Soup, a data-free variant of the original method that uses geometric alignment as a selection criterion, showing that alignment is a good proxy for merging effectiveness.

3. We introduce MonoSoup, a data-free, post-hoc editing approach that improves the ID-OOD trade-off using only a single fine-tuned model, which typically suffers from degraded OOD performance on its own due to representation collapse. Our method decomposes each layer's update into high- and low-energy components and adaptively reweights them, eliminating the need for multiple checkpoints.

4. We empirically validate our approach on vision (CLIP) and language (Qwen) benchmarks, demonstrating that MonoSoup consistently improves OOD generalization while maintaining or enhancing in-distribution accuracy.

## 2 PRELIMINARIES

**Model Soups** The common practice in machine learning is to select the single best model from a hyperparameter search for final deployment, based on a validation metric, and discard the remaining $m - 1$ checkpoints. However, models originating from the same pre-trained initialization often occupy a connected, low-loss basin in the optimization landscape (Garipov et al., 2018; Frankle et al., 2020; Izmailov et al., 2018), making them amenable to ensembling. Formally, consider $m$ weights $\{\theta_t\}_{t \in [m]}$, obtained by fine-tuning the pre-trained weights $\theta_0$ on a target dataset $\mathcal{D}_{\text{train}}$ with $m$ different hyperparameter configurations. Model Soups (Wortsman et al., 2022a) leverages this insight by averaging the weights of multiple fine-tuned models, resulting in the final parameters

$\boldsymbol{\theta} = \frac{1}{T}\sum_{t=1}^{T}\boldsymbol{\theta}_t$. While effective, soups are computationally expensive: their benefits are most pronounced when averaging many diverse checkpoints, which is impractical for large-scale models.

**Model Stock** In an effort to reduce the significant computational and storage overhead of Model Soups, Model Stock (Jang et al., 2024) requires only two models and operates layer-wise, based on the idea that cosine similarity can quantify the signal-to-noise ratio of the fine-tuning updates. Let $\boldsymbol{W}_0^{(\ell)}$ denote the pre-trained weights at layer $\ell$, and $\boldsymbol{W}_1^{(\ell)}, \boldsymbol{W}_2^{(\ell)}$ the corresponding updates from the two checkpoints. Model Stock first computes the cosine similarity $\cos\alpha^{(\ell)}$ between these task vectors to measure their agreement. The merged weights are:

$$\boldsymbol{W}_{\text{stock}}^{(\ell)} = \boldsymbol{W}_0^{(\ell)} + \lambda^{(\ell)} \cdot \left(\frac{\boldsymbol{W}_1^{(\ell)} + \boldsymbol{W}_2^{(\ell)}}{2}\right), \text{ where } \lambda^{(\ell)} = \frac{2\cos\alpha^{(\ell)}}{1 + \cos\alpha^{(\ell)}}. \tag{1}$$

This rule preserves directions where the updates are well-aligned ($\cos\alpha^{(\ell)} \to 1$), while reverting toward the pre-trained weights when they disagree. Compared to soups, it is more efficient since it reduces the number of required models from a large ensemble to just two, but still assumes access to at least two fine-tuned checkpoints.

**Wise-FT and LiNeS** Beyond multi-checkpoint methods, there also exist approaches that operate with a single fine-tuned model. Wise-FT (Wortsman et al., 2022b) improves generalization by linearly interpolating between the pre-trained and fine-tuned weights. Given a coefficient $\lambda \in [0, 1]$, the merged parameters are $\boldsymbol{\theta}_{\text{wise}} = (1 - \lambda)\boldsymbol{\theta}_0 + \lambda\boldsymbol{\theta}_t$, which traces a continuum between robustness (closer to $\boldsymbol{\theta}_0$) and specialization (closer to $\boldsymbol{\theta}_t$). Therefore, Wise-FT delivers a family of models along this path rather than a single edited checkpoint, and it applies the same interpolation uniformly across all layers and directions, limiting its ability to capture anisotropic updates. LiNeS (Wang et al., 2025) also operates on a single checkpoint, introducing post-training layer scaling to prevent catastrophic forgetting (McCloskey & Cohen, 1989) and enhance model merging. However, it requires labeled data to tune its hyperparameters and employs a single coefficient for all layers within a transformer block, potentially overlooking the distinct dynamics of linear and attention layers.

## 3 THE ROLE OF ALIGNMENT IN MODEL MERGING

In this section, we investigate the conditions under which multi-model merging succeeds, finding that success often depends on the alignment of the fine-tuning updates. To investigate this, we use Model Stock (Jang et al., 2024) as a probe: since its layer-wise weighting rule explicitly depends on cosine similarity as in Equation 1, it provides a natural lens to study how alignment relates to merging performance.

We evaluate all pairwise combinations among 70 CLIP ViT-B/32 models (Wortsman et al., 2022a) fine-tuned on ImageNet, each model corressponds to a different hyper-parameter configuration. We compare each merged model against the better of its two constituents across five natural distribution shifts: ImageNet-V2 (Recht et al., 2019), ImageNet-R (Hendrycks et al., 2021a), ImageNet-Sketch (Wang et al., 2019), ImageNet-A (Hendrycks et al., 2021c), and ObjectNet (Barbu et al., 2019). Specifically, we define the performance differences of model stock w.r.t. the involved models:

$$\Delta\text{acc}^{\text{ID}} = \text{acc}_{\text{MS}}^{\text{ID}} - \max\left\{\text{acc}_1^{\text{ID}}, \text{acc}_2^{\text{ID}}\right\}$$

$$\Delta\text{acc}^{\text{OOD}} = \text{acc}_{\text{MS}}^{\text{OOD}} - \max\left\{\text{acc}_1^{\text{OOD}}, \text{acc}_2^{\text{OOD}}\right\}$$

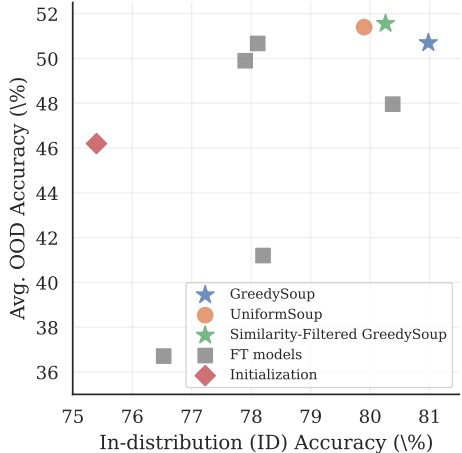

Figure 1: Performance of Similarity-Filtered Greedy Soup (SFGS) on CLIP ViT-B/32 checkpoints. SFGS achieves competitive ID and OOD performance relative to validation-based greedy soup and the strongest single checkpoint, supporting the observation that alignment is a key indicator of merging effectiveness.

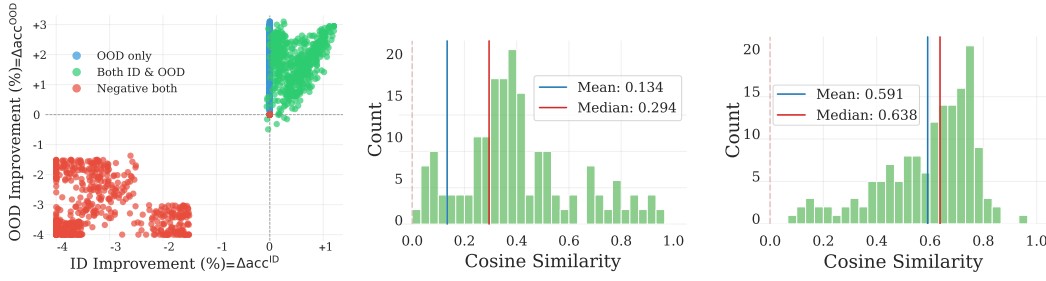

(a) Performance scatter.  (b) Low-performing pair similarity.  (c) High-performing pair similarity.

Figure 2: Performance and alignment analysis of Model Stock on 2,409 pairwise combinations of CLIP ViT-B/32 models fine-tuned on ImageNet. (a) Scatter plot of ID vs. OOD performance relative to the better constituent model. (b) and (c): Layer-wise cosine similarity for low-performing and high-performing, respectively. Stronger alignment coincides with consistent gains, highlighting that alignment can serve as a key predictor of merging success.

As illustrated in Figure 2a, Model Stock performance highly relies on the pair selection. We randomly select a pair from each mode, showing a histogram of per-layer cosine similarities for a low- and high-performing pair in Figure 2b and Figure 2c, respectively. We observe performance improvements when task vectors are well aligned, but its benefits diminish when they are weakly correlated. This sensitivity highlights a key principle: merging is effective when the fine-tuning updates are well-aligned, but fails when conflicting updates interfere.

To validate this observation, we tested a simple variant of Greedy Soup, which replaces validation-based selection with a geometric filter. Starting with the best ID-performing model, we include a candidate checkpoint if its average layer-wise cosine similarity to the current soup exceeds a threshold $\delta$: $\frac{1}{|S|} \sum_{j \in S} \cos(\tau_i, \tau_j) \geq \delta$. As shown in Figure 1, this lightweight procedure, termed *Similarity-Filtered Greedy Soup*, achieves performance comparable to validation-based greedy soup (Wortsman et al., 2022a), showing that geometric alignment might serve as a reliable proxy for effective merging. Nevertheless, like all soup-based methods, it still requires multiple fine-tuned checkpoints. Taken together, these analyses suggest that successful merging of models originating from the same pre-trained initialization hinges on reinforcing well-aligned directions while suppressing noisy or conflicting ones. In the next section, we explore if these principles apply within a single model: its fine-tuning update may contain both dominant, task-relevant directions as well as weaker components that can harm generalization.

## 4 MONOSOUP

Our analysis of multi-model merging suggests that performance gains arise when fine-tuning updates reinforce shared directions while suppressing noisy or conflicting ones. This motivates searching for analogous signals *within a single checkpoint*. Specifically, we hypothesize that the update of a fine-tuned model contains both dominant directions that capture task-specific adaptation and weaker components that, while less prominent, are important for maintaining generalization.

To make this structure explicit, we analyze the weight difference matrix at each layer $W^{(\ell)} = W_1^{(\ell)} - W_0^{(\ell)} \in \mathbb{R}^{m \times n}$, where $W_0^{(\ell)}$ and $W_1^{(\ell)}$ are the pre-trained and fine-tuned weights for layer $\ell$, respectively. Applying singular value decomposition (SVD), $W^{(\ell)} = U^{(\ell)} \Sigma^{(\ell)} V^{(\ell)\top}$, where the spectrum of singular values $\sigma_1 \geq \sigma_2 \geq \dots$ quantifies how the adaptation is distributed across directions. We partition this spectrum into two components:

$$W_{\text{High}}^{(\ell)} = \sum_{i \leq k} \sigma_i^{(\ell)} u_i^{(\ell)} v_i^{(\ell)\top}, \qquad W_{\text{Low}}^{(\ell)} = W^{(\ell)} - W_{\text{High}}^{(\ell)}, \qquad (2)$$

where $k$ is the smallest index that preserves at least a fraction $R$ of the spectral energy:

$$k = \operatorname*{argmin}_{j} \left\{ j \;\middle|\; \frac{\sum_{i=1}^{j} \sigma_i^2}{\sum_{i=1}^{\min(m,n)} \sigma_i^2} \geq R \right\}. \qquad (3)$$

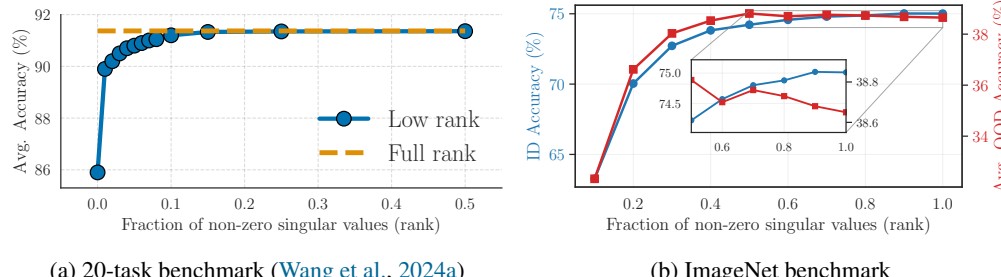

(a) 20-task benchmark (Wang et al., 2024a)   (b) ImageNet benchmark

Figure 3: Effect of truncating low-energy components on different benchmarks. (a) On the 20-task vision benchmark, performance saturates after retaining only a small number of singular values, consistent with prior reports that low-rank updates suffice. (b) On ImageNet with natural OOD shifts, truncation substantially reduces both ID and OOD accuracy, even when preserving 95% of spectral energy. This highlights that, in large-scale fine-tuning, low-energy directions carry critical information for generalization and cannot simply be removed. See Appendix D for further details.

Intuitively, the high-energy spectral component $W_{\text{High}}^{(\ell)}$ encodes concentrated task-specific adaptation, while the low-energy $W_{\text{Low}}^{(\ell)}$ contains residual updates that, despite potentially capturing noise, may preserve information critical for OOD robustness. A natural question is whether the low-energy component can be removed altogether. Several recent studies (Gargiulo et al., 2025; Tang et al., 2025; Stoica et al., 2024) have argued that $W_{\text{Low}}^{(\ell)}$ largely encodes noise and that discarding it can improve merging. These results, however, are mostly based on the standard task arithmetic benchmark (Ilharco et al., 2023), where the CLIP vision encoder is fine-tuned on a collection of relatively small-scale classification tasks. In Figure 3a, we progressively remove a larger amount of singular values and track the average performance on the suite of 20 tasks proposed by Wang et al. (2024a). In this regime, adaptation tends to be concentrated in a few dominant directions, so truncation appears effective.

In contrast, our setting involves fine-tuning on ImageNet and evaluation across natural distribution shifts (ImageNetV2, ImageNet-R, ImageNet-Sketch, ObjectNet, ImageNet-A). As shown in Figure 3b, removing low-energy components in this regime leads to degradation of both ID and OOD accuracy, even when retaining 95% of spectral energy. This suggests that low-energy directions encode complementary information that is essential for OOD robustness.

To balance specialization and generalization, we reweigh the high- and low-energy components rather than discarding one of them. For each layer, the edited update is

$$W_{\text{MonoSoup}}^{(\ell)} = \lambda_{\text{High}}^{(\ell)} W_{\text{High}}^{(\ell)} + \lambda_{\text{Low}}^{(\ell)} W_{\text{Low}}^{(\ell)}, \tag{4}$$

where coefficients $\lambda_{\text{High}}^{(\ell)}$ and $\lambda_{\text{Low}}^{(\ell)}$ are determined adaptively. We now turn to how low-energy directions should be weighted relative to the dominant ones. We rely on two complementary signals derived directly from the model. The first comes from the singular value spectrum itself: when the decay is steep, most adaptation is captured by the leading singular values, suggesting that residual directions are less informative. When the singular value spectrum is flat, however, the contribution of weaker directions is more substantial. To capture this behavior, we define a spectral decay ratio,

$$\rho^{(\ell)} = \left( \frac{\sigma_{k+1}(W^{(\ell)})}{\sigma_1(W^{(\ell)})} \right)^2, \tag{5}$$

which is small when the spectrum decays steeply and large when it is flat.

The second signal quantifies how much of the fine-tuning update lies in low-energy directions. Let $W = \Delta W^{(\ell)}$ and decompose it as $W = W_{\text{High}}^{(\ell)} + W_{\text{Low}}^{(\ell)}$ as in Equation 2. We define

$$\cos^2 \alpha^{(\ell)} = \frac{\left\| W_{\text{Low}}^{(\ell)} \right\|_F^2}{\left\| W^{(\ell)} \right\|_F^2} \in [0, 1]$$

so that $\cos \alpha^{(\ell)}$ measures the fraction of update energy carried by low-energy directions (see Appendix C). Larger $\cos \alpha^{(\ell)}$ indicates that the fine-tuning update is spread over many weak directions rather than being concentrated in a small number of dominant singular vectors. As we show in Appendix Appendix H using a CKA analysis of hidden representations, these low-energy directions are precisely the ones that preserve pretrained features on OOD inputs while providing only mild specialization on ID.

The final coefficients are given by

$$\lambda_{\text{Low}}^{(\ell)} = \rho^{(\ell)} + \left(1 - \rho^{(\ell)}\right) \cos \alpha^{(\ell)}, \qquad \lambda_{\text{High}}^{(\ell)} = 1 - \lambda_{\text{Low}}^{(\ell)}. \tag{6}$$

Thus $\lambda_{\text{Low}}^{(\ell)} = \rho^{(\ell)} + \left(1 - \rho^{(\ell)}\right) \cos \alpha^{(\ell)}$ increases when both (i) the spectrum is relatively flat and (ii) a substantial fraction of update energy lies in low-energy directions, and these are exactly the cases where re-emphasizing $W_{\text{low}}$ improves OOD robustness while keeping ID strong. This form ensures consistent behavior in the natural limits: when $\rho^{(\ell)} \to 0$ and $\cos \alpha^{(\ell)} \to 0$, we obtain $\lambda_{\text{Low}}^{(\ell)} \to 0$ and suppress residual directions; when $\rho^{(\ell)} \to 1$ or $\cos \alpha^{(\ell)} \to 1$, then $\lambda_{\text{Low}}^{(\ell)} \to 1$ and all low-energy directions are retained. Intuitively, $\rho^{(\ell)}$ provides a baseline estimate of how much residual mass to keep, while $\cos \alpha^{(\ell)}$ restores low-energy directions when they align with the pre-trained model. The only hyperparameter is the energy threshold $R$ in Equation 3, which is directly interpretable as the fraction of spectral energy used to define the high-energy component.

## 5 EXPERIMENTS

We next evaluate MonoSoup across both vision and language domains. On CLIP models, we compare against prior merging methods such as Model Soups and ModelStock, testing whether MonoSoup can achieve competitive or superior robustness using only a single fine-tuned checkpoint. We then extend the evaluation to large language models from the Qwen family (Yang et al., 2025), where we assess its effectiveness on mathematical reasoning and multiple-choice benchmarks. Finally, we study its integration with Wise-FT to examine complementarity with interpolation-based robustness methods.

Table 1: Comparison of merging methods on CLIP ViT-B/32 fine-tuned on ImageNet. We report top-1 accuracy on ImageNet (ID) and average across five OOD shifts. $+$ refers to the best-performing model on each metric (ID or OOD), while $-$ refers to the worst-performing model. Cost refers to the number of used checkpoints. Uniform and Greedy Soups require up to 70 checkpoints, and ModelStock requires two. In contrast, MonoSoup matches or surpasses them with a single checkpoint, improving Avg. OOD from 50.67% to **51.60%** on the best-OOD model and adding almost **+8%** on weaker ones.

| Method | ID | Avg. OOD | Cost |
|---|---|---|---|
| Initialization | 75.4% | 46.2% | |
| FT model (OOD$^+$) | 78.11% | 50.67% | |
| FT model (OOD$^-$) | 76.53% | 36.71% | |
| FT model (ID$^+$) | 80.38% | 47.96% | |
| FT model (ID$^-$) | 74.99% | 38.64% | |
| *Prior Soups-Merging Methods* | | | |
| Uniform Model Soup | 79.9% | 51.4% | 70 |
| Greedy Model Soup | **81.0%** | 50.7% | 70 |
| *ModelStock (Pairwise Models)* | | | |
| ID$^+$, OOD$^+$ | 79.39% | 50.53% | 2 |
| ID$^+$, OOD$^-$ | 78.43% | 49.39% | 2 |
| ID$^+$, ID$^-$ | 78.32% | 50.63% | 2 |
| OOD$^+$, OOD$^-$ | 76.76% | 48.41% | 2 |
| OOD$^+$, ID$^-$ | 77.49% | 51.02% | 2 |
| ID$^-$, OOD$^-$ | 78.09% | 47.81% | 2 |
| *MonoSoup (Single Models)* | | | |
| OOD$^+$ | 78.29% | **51.60%** | 1 |
| OOD$^-$ | 78.55% | 44.21% | 1 |
| ID$^+$ | 80.03% | 49.95% | 1 |
| ID$^-$ | 77.76% | 46.54% | 1 |
| *MonoSoup (Pairwise Models)* | | | |
| ID$^+$, OOD$^+$ | 80.10% | 51.37% | 2 |
| ID$^+$, OOD$^-$ | 78.87% | 48.37% | 2 |
| ID$^+$, ID$^-$ | 79.02% | 50.12% | 2 |
| OOD$^+$, OOD$^-$ | 78.44% | 49.26% | 2 |
| OOD$^+$, ID$^-$ | 78.05% | 50.48% | 2 |
| ID$^-$, OOD$^-$ | 78.94% | 47.79% | 2 |

### 5.1 MERGING VISION TRANSFORMERS

We begin with CLIP ViT-B/32 models fine-tuned on ImageNet, the standard testbed for soup-based approaches. In-distribution (ID) accuracy is measured on ImageNet-1K, while out-of-distribution (OOD) robustness is assessed on five natural shifts: ImageNet-V2, ImageNet-R, ImageNet-Sketch, ImageNet-A, and ObjectNet. We use the 70 CLIP ViT-B/32 checkpoints released by Wortsman et al. (2022a). Since presenting results for all 70 models would be impractical, we focus on four

Table 2: Evaluation of Qwen3-0.6B across mathematical reasoning and multiple-choice benchmarks. MonoSoup consistently improves performance over the fine-tuned model and LiNeS, with the largest gains on the most challenging tasks of $GSM_{Plus}$ and $GSM8K_{Platinum}$. MonoSoup achieves comparable or better performance to ModelStock using only a single checkpoint.

| Method | GSM8K | $GSM_{Plus}$ | $GSM8K_{Platinum}$ | SciQ | MMLU-Pro-Math |
|---|---|---|---|---|---|
| QWEN3-0.6B-BASE | 52.6 | 22.5 | 50.1 | 92.6 | 33.7 |
| *Linear Learning Rate Variants (M-1)* | | | | | |
| M-1 | 55.8 | 29.5 | 55.6 | 94.6 | 35.6 |
| M-1 + LINES | 56.2 | 30.1 | 56.3 | 94.9 | 36.7 |
| M-1 + MONOSOUP | **56.7** | 30.3 | 56.7 | **95.1** | **37.2** |
| *Cosine Learning Rate Variants (M-2)* | | | | | |
| M-2 | 56.1 | 30.8 | 58.5 | 94.5 | 35.9 |
| M-2 + LINES | 56.5 | 31.4 | 58.9 | 95.2 | 36.1 |
| M-2 + MONOSOUP | 56.6 | **31.7** | **59.3** | 95.1 | 36.6 |
| *Cosine Learning Rate with Extended Training (M-3)* | | | | | |
| M-3 | 55.9 | 30.6 | 57.3 | 93.5 | 34.8 |
| M-3 + LINES | 56.3 | 30.8 | 58.1 | 93.8 | 35.2 |
| M-3 + MONOSOUP | 56.6 | 31.4 | 58.8 | 94.2 | 35.5 |
| *ModelStock Merging Method* | | | | | |
| (M-1, M-2) | 56.6 | 31.5 | 59.0 | 94.9 | 36.8 |
| (M-1, M-3) | 56.4 | 31.3 | 58.7 | 94.8 | 36.4 |
| (M-2, M-3) | 56.5 | 31.6 | 58.9 | 95.2 | 36.7 |

representative cases: the checkpoint with the highest ID accuracy ($ID^+$), the lowest ID accuracy ($ID^-$), the highest OOD accuracy ($OOD^+$), and the lowest OOD accuracy ($OOD^-$). This selection allows us to illustrate how MonoSoup behaves across both strong and weak checkpoints. Unless otherwise stated, we set $R = 0.8$ for MonoSoup. We ablate this hyperparameter in subsection 5.4; for intuition about the connection between $R$ and $\cos \alpha$, see Appendix C.

Table 1 presents the results for vision transformers. We also report the number of checkpoints required by each method in the *Cost* column. MonoSoup consistently matches or surpasses multi-model approaches while requiring only a single checkpoint. On the strongest OOD model, Avg. OOD increases from 50.67% to 51.60%, surpassing Greedy Soup without aggregating 70 models. On weaker checkpoints, MonoSoup recovers collapsed representations, improving Avg. OOD by +7.9% for the worst-ID model and +7.5% for the worst-OOD model. When applied to pairs of fine-tuned models, MonoSoup achieves a better balance between ID and OOD performance compared to ModelStock. This is especially pronounced in the poor of ($OOD^+$) where MonoSoup dominates on both objectives. This shows that the benefits of the proposed method extend beyond the single-checkpoint setting.

## 5.2 MERGING LARGE LANGUAGE MODELS

To test the generality of MonoSoup beyond vision, we evaluate it on large language models. Specifically, we fine-tune multiple variants of the Qwen3-0.6B model, each with different hyperparameter configurations, such as learning rates, training epochs, and schedules. Further details are provided in Appendix E. All variants are trained on a mixture spanning mathematical reasoning and multiple-choice question answering: MetaMathQA (Yu et al., 2023), which augments GSM8K (Cobbe et al., 2021) and MATH (Hendrycks et al., 2021b); DeepMind-AquaRat (Ling et al., 2017); and the multiple-choice datasets OpenBookQA (Mihaylov et al., 2018) and SciQ (Welbl et al., 2017).

For evaluation, we propose a benchmark that spans a wide spectrum of reasoning difficulty. It includes GSM8K (Cobbe et al., 2021) and SciQ, which overlap with the training mixture and are treated as in-distribution tasks, as well as $GSM_{Plus}$ (Li et al., 2024), $GSM8K_{Platinum}$ (Vendrow et al., 2025), and MMLU-Pro-Math (Wang et al., 2024b), which probe advanced or adversarial reasoning skills not explicitly covered during training and thus serve as out-of-distribution evaluations. While this ID/OOD split is less rigid than in vision benchmarks such as CLIP, the increasing task difficulty provides an analogous way to assess robustness and generalization in the language domain.

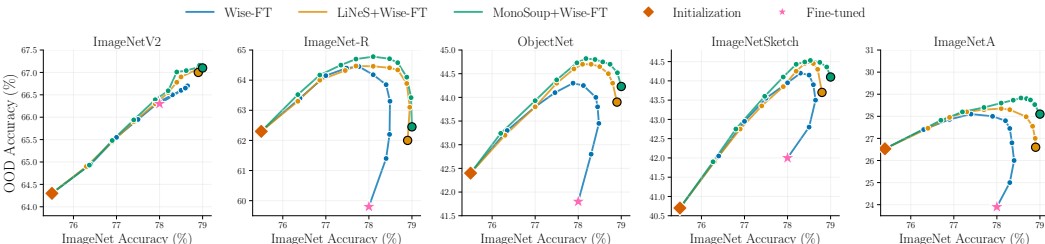

Figure 4: MonoSoup integrated with Wise-FT on CLIP ViT-B/32. MonoSoup improves ID and OOD accuracy across individual checkpoints. When combined with Wise-FT, the Pareto fronts consistently dominate those of Wise-FT and LiNeS, showing that MonoSoup provides a stronger endpoint for interpolation-based robustness.

The results are presented in Table 2 and demonstrate consistent improvements across all fine-tuned variants. MonoSoup improves over the baseline Qwen3-0.6B models and surpasses LiNeS across every benchmark, with the largest gains observed on GSM$_{Plus}$ and GSM8K$_{Platinum}$ (+9.2 points each). Compared to ModelStock, which remains competitive when merging pairs of models, MonoSoup matches or exceeds its performance while requiring only a single checkpoint. These findings mirror the results on CLIP: MonoSoup enhances robustness and generalization without reliance on ensembles, scaling naturally when multiple models are available but remaining highly effective in the single-checkpoint setting.

## 5.3 Integration with Wise-FT

We next study how MonoSoup interacts with Wise-FT (Wortsman et al., 2022b), which interpolates between pre-trained and fine-tuned weights to produce a continuum of models that trace the trade-off between ID and OOD accuracy. This setting allows us to test whether MonoSoup can serve as a stronger endpoint for interpolation-based robustness methods. We also compare against LiNeS (Wang et al., 2025), a post-training technique that linearly scales fine-tuning updates according to layer depth. However, using a single coefficient per transformer block neglects the different fine-tuning dynamics among parameter groups, such as attention versus feedforward layers (Yang et al., 2024). In contrast, MonoSoup is fully data-free and adapts coefficients at the level of individual subspaces within each layer.

We use the same experimental settings as the vision experiments. We report the average results over all 70 released checkpoints in Figure 4. We observe that applying MonoSoup improves both ID and OOD performance, even without interpolation. When combined with Wise-FT, the resulting Pareto fronts consistently dominate those of Wise-FT alone, while also surpassing LiNeS. This demonstrates that MonoSoup is complementary to other techniques: it enhances a single checkpoint in a data-free way, and this stronger base further amplifies the benefits of existing robustness techniques.

## 5.4 Analysis and Discussion

We finally analyze two design aspects of MonoSoup: the effect of the spectral energy threshold $R$ and the role of each component in the mixing rule. Varying $R$ controls how much of the fine-tuning update is assigned to the high-energy subspace. The results on CLIP ViT-L/14, shown in the left panel of Figure 5, reveal three clear patterns. When $R$ is too small, too much spectral mass is discarded, which hurts both ID and OOD performance. Very large values of $R$ keep almost all directions, saturating improvements and sometimes leading to collapse. Intermediate values around 0.7–0.85 achieve the best balance, confirming that low-energy directions are important but must be modulated relative to dominant task-specific updates.

We also ablate the two signals in our mixing rule: spectral decay $\rho^\ell$ and alignment $\cos\alpha^\ell$. The right panel of Figure 5 shows that relying only on $\rho^\ell$ preserves ID accuracy but brings little OOD improvement, while relying only on $\cos\alpha^\ell$ improves OOD on weaker checkpoints but can reduce ID. Uniform mixing gives inconsistent results, and keeping only the high- or low-energy components degrades one side of the trade-off. In contrast, combining $\rho^\ell$ and $\cos\alpha^\ell$ consistently

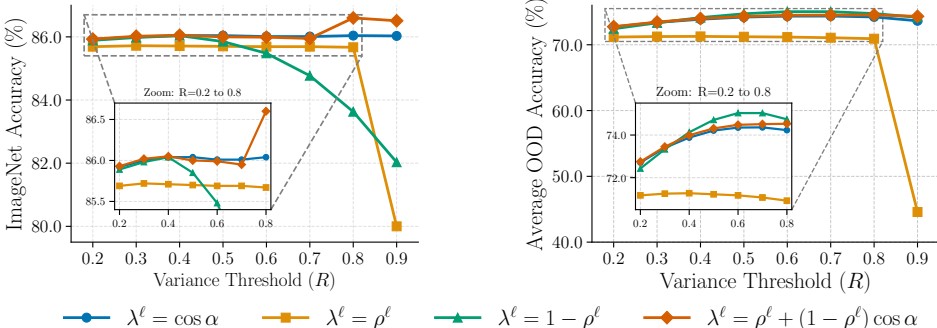

Figure 5: **Component Analysis.** Effect of varying the variance threshold $R$ and the contributions of each term in the coefficient $\lambda^\ell$ on CLIP ViT-L/14. Results are stable across a wide range of $R$ values, and both the spectral decay and cosine overlap components contribute meaningfully to the final balance between ID and OOD performance.

yields the strongest OOD performance without sacrificing ID, and remains stable across a wide range of $R$. These results validate the design of MonoSoup: both spectral decay and alignment provide complementary signals, and together they enable a principled way to retain the benefits of low-energy directions without undermining task-specific adaptations.

## 6  RELATED WORK

**Representation Collapse and Robust Fine-Tuning.** The prevalent pre-train-then-finetune paradigm often leads to a degradation of a model's general-purpose knowledge, resulting in a decline in out-of-distribution (OOD) performance (Kumar et al., 2022). This phenomenon, termed *representation collapse* (Aghajanyan et al., 2020), has motivated a significant body of research focused on making the fine-tuning process more robust. Such methods typically regularize the fine-tuning process to preserve the valuable features learned during pre-training, thereby improving OOD generalization (Gouk et al., 2021; Zhang et al.; Razdaibiedina et al., 2023; Lee et al., 2023; Goyal et al., 2023; Wortsman et al., 2022b; Mao et al., 2023; Nam et al., 2024; Oh et al., 2024). While effective, these approaches intervene directly in the computationally expensive fine-tuning stage, motivating the exploration of more efficient, post-hoc alternatives.

**Mode Connectivity and Post-hoc Merging.** An alternative line of work focuses on post-hoc manipulation of model weights, a practice theoretically grounded in the properties of the neural network loss landscape. Seminal works showed that distinct solutions found by separate training runs can be connected by a non-linear path of low loss (Garipov et al., 2018; Draxler et al., 2018). More critically for fine-tuning, Frankle et al. (2020) demonstrated the existence of *linear mode connectivity* between models that share the same pre-trained initialization. This property enables simple yet powerful techniques like weight averaging. By interpolating the parameters of multiple fine-tuned checkpoints, these methods have been shown to find wider, more robust optima (Izmailov et al., 2018; Wortsman et al., 2021), leading to improved in-distribution (Wortsman et al., 2022a; Jang et al., 2024) and out-of-distribution performance (Wortsman et al., 2022b; Ramé et al.; Rame et al., 2023) without requiring additional inference costs.

**Unifying Principles of Successful Merging.** Recent analyses of these merging techniques have revealed a unifying principle: successful weight-space ensembling reinforces dominant directions in the weight space that encode shared, task-relevant signals, while simultaneously suppressing noisy or misaligned directions that harm generalization (Wortsman et al., 2022a; Jang et al., 2024; Rame et al., 2023). This insight has inspired the development of more sophisticated merging strategies that explicitly identify and manipulate these core components of the fine-tuning update (Gargiulo et al., 2025; Tang et al., 2025; Wang et al., 2025). Beyond improving single-model robustness, these principles of weight-space arithmetic have been successfully extended to a broader range of applications, including multi-task learning (Ilharco et al., 2022; 2023; Dimitriadis et al., 2023; Yadav et al., 2023) and multi-objective alignment (Ramé et al., 2024; Zhong et al., 2024).

## 7 CONCLUSION

In this paper, we introduced MonoSoup, a data-free method that reweights the spectral components of fine-tuning updates to improve both in-distribution accuracy and out-of-distribution robustness from a single checkpoint. Unlike prior approaches that depend on ensembles of fine-tuned models or carefully aligned pairs, MonoSoup compresses their benefits into a lightweight, single-model procedure that restores OOD performance even for weak checkpoints. Experiments on CLIP and Qwen benchmarks show its effectiveness across vision and language domains, demonstrating that robust gains are possible without the computational and storage overhead of multi-model methods. A current limitation is the reliance on a variance-retention threshold $R$, which, although interpretable, introduces a hyperparameter that may require adaptation across architectures or domains. Looking forward, extending our approach beyond vision and language to other modalities offers a promising direction. Overall, our results highlight that the benefits of model soups can be retained—even strengthened—without the burden of maintaining large ensembles, making MonoSoup a practical plug-and-play tool for reliable model deployment.

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

Table 3: A comprehensive comparison of single-model merging methods on CLIP ViT-B/32. Performance metrics are presented for ImageNet and the average of five OOD datasets across fine-tuned models utilizing *LP initialization* released by Wortsman et al. (2022a). For Wise-FT, we sweep the interpolation coefficient $\alpha$ and report the best-performing setting with respect to Avg. OOD.

| Method | Linear Probe initialization | |
| --- | --- | --- |
| | ID (ImageNet) | Avg. OOD |
| *Baseline* | | |
| CLIP LP Initialization | 75.40% | 46.20% |
| *Fine-Tuned Models* | | |
| FT model (Best Avg. OOD) | 78.11% | 50.67% |
| FT model (Worst Avg. OOD) | 76.53% | 36.71% |
| FT model (Best ID) | 80.38% | 47.96% |
| FT model (Worst ID) | 74.99% | 38.64% |
| *LiNeS* | | |
| LiNeS+FT model (Best Avg. OOD) | 78.25% | 51.56% |
| LiNeS+FT model (Worst Avg. OOD) | 78.13% | 41.33% |
| LiNeS+FT model (Best ID) | 80.46% | 49.10% |
| LiNeS+FT model (Worst ID) | 77.31% | 44.96% |
| *Wise-FT* | | |
| Wise-FT+FT model (Best Avg. OOD) | 78.12% | 51.54% |
| Wise-FT+FT model (Worst Avg. OOD) | 77.11% | 45.2% |
| Wise-FT+FT model (Best ID) | 78.73% | 49.12% |
| Wise-FT+FT model (Worst ID) | 78.00% | 45.75% |
| *Our Proposed Method* | | |
| MonoSoup +FT model (Best Avg. OOD) | 78.29% | 51.60% |
| MonoSoup +FT model (Worst Avg. OOD) | 78.55% | 44.21% |
| MonoSoup +FT model (Best ID) | 80.03% | 49.95% |
| MonoSoup +FT model (Worst ID) | 77.76% | 46.54% |

## A  COMPARISON WITH SINGLE-MODEL MERGING METHODS

**Linear Probing initialization (LP init).** Table 3 presents a comprehensive analysis of single-model merging methods for models with *Linear Probing initialization (LP init)*. MonoSoup significantly improves the M-14 model, which has the worst average OOD performance, increasing OOD accuracy from 36.71% to 44.21% (a gain of 7.5%) and ID accuracy by 2.02%. Furthermore, it enhances the M-31 model, which has the worst ID performance, achieving a 2.77% accuracy gain.

## B  ZERO-SHOT INITIALIZATION (ZS INIT)

Table 4 presents a comprehensive analysis of single-model merging methods for models with *zero-shot initialization (ZS init)*. We use two publicly available *ZS*-initialized checkpoints from Jang et al. (2024). Our proposed MonoSoup demonstrates achieves improvements of 6.4% in average OOD accuracy and 0.8% and 0.6% in ID accuracy across the respective experimental configurations. For comparison with *soup*-model merging methods in the two-checkpoint scenario, we apply our method to the average of two checkpoints, resulting in performance gains of 2.9% over ModelStock, 3.0% over GreedySoup, and 1.8% over UniformSoups.

## C  CONNECTION BETWEEN $R$ AND $\cos\alpha$

Let $r = \text{rank}(\boldsymbol{W})$ and let $\sigma_j = \sigma_j(\boldsymbol{W})$ denote the $j$-th singular value of $\boldsymbol{W}$ for $j \in \{1, \ldots, r\}$. The Frobenius norm of $\boldsymbol{W}$ can be expressed in terms of its singular values as

$$\|\boldsymbol{W}\|_F^2 = \sum_{j=1}^{r} \sigma_j^2.$$

Table 4: Comparison of single-model merging methods on CLIP ViT-B/32 with *zero-shot initialization (ZS init)*. We use the two publicly available checkpoints from Jang et al. (2024) and use their reported numbers for the Model Soups baselines with 48 models.

| Method | Zero-Shot Initialization | | |
|---|---|---|---|
| | ID (ImageNet) | Avg. OOD | Cost |
| *Baselines* | | | |
| Initialization | 63.3% | 48.5% | 0 |
| Vanilla FT 1 | 78.1% | 46.7% | 1 |
| Vanilla FT 2 | 78.3% | 46.9% | 1 |
| *LiNeS* | | | |
| LiNeS ($\alpha$=0.1, $\beta$=0.9) + FT 1 | 78.7% | 52.2% | 1 |
| LiNeS ($\alpha$=0.5, $\beta$=0.5) + FT 1 | 78.9% | 51.1% | 1 |
| LiNeS ($\alpha$=0.1, $\beta$=0.9) + FT 2 | 78.5% | 51.9% | 1 |
| LiNeS ($\alpha$=0.5, $\beta$=0.5) + FT 2 | 79.0% | 51.1% | 1 |
| *Wise-FT* | | | |
| Wise-FT+ FT 1 | 78.8% | 52.5% | 1 |
| Wise-FT+ FT 2 | 78.8% | 52.6% | 1 |
| *Prior Soups-Merging Methods* | | | |
| Uniform Model Soup | 79.7% | 52.0% | 48 |
| Greedy Model Soup | 80.4% | 50.8% | 48 |
| ModelStock | 79.8% | 50.9% | 2 |
| *Our Proposed Method* | | | |
| MonoSoup w/ Vanilla FT 1 | 78.9% | 53.1% | 1 |
| MonoSoup w/ Vanilla FT 2 | 78.9% | 53.2% | 1 |
| MonoSoup w/ Avg. of FT 1&2 | 79.0% | 53.8% | 2 |

Given a target variance capture ratio $R \in [0, 1]$, we define the truncation index $k$ as

$$k = \underset{j \in \{1,\dots,r\}}{\arg\min} \left\{ j \,\middle|\, \frac{\sum_{s=1}^{j} \sigma_s^2}{\|\boldsymbol{W}\|_F^2} \geq R \right\}. \tag{7}$$

Let $P_k = \sum_{s=1}^{k} \sigma_s^2 / \|\boldsymbol{W}\|_F^2$ denote the actual fraction of variance captured by the truncated matrix $\boldsymbol{W}_{\text{High-Energy}}$. By the definition of $k$, we have

$$P_k \geq R.$$

**Lemma 1.** *If $k > 1$, then*

$$P_k - \frac{\sigma_k^2}{\|\boldsymbol{W}\|_F^2} < R \leq P_k.$$

*Proof.* Since $k$ is the minimum index satisfying the variance threshold, we have that $k - 1$ does not satisfy it, i.e.,

$$\frac{\sum_{s=1}^{k-1} \sigma_s^2}{\|\boldsymbol{W}\|_F^2} < R.$$

Observing that $\sum_{s=1}^{k-1} \sigma_s^2 = \sum_{s=1}^{k} \sigma_s^2 - \sigma_k^2$, we obtain

$$P_k - \frac{\sigma_k^2}{\|\boldsymbol{W}\|_F^2} < R.$$

Combined with $P_k \geq R$, the result follows. $\square$

Now, we establish the relationship with $\cos \alpha$. By the orthogonal decomposition $\boldsymbol{W} = \boldsymbol{W}_{\text{High-Energy}} + \boldsymbol{W}_{\text{Low-Energy}}$ and the Pythagorean theorem in Frobenius norm, we have

$$\|\boldsymbol{W}\|_F^2 = \|\boldsymbol{W}_{\text{High-Energy}}\|_F^2 + \|\boldsymbol{W}_{\text{Low-Energy}}\|_F^2.$$

Since $\cos \alpha = \|\boldsymbol{W}_{\text{Low-Energy}}\|_F / \|\boldsymbol{W}\|_F$, we obtain

$$\cos^2 \alpha = \frac{\|\boldsymbol{W}_{\text{Low-Energy}}\|_F^2}{\|\boldsymbol{W}\|_F^2} = \frac{\|\boldsymbol{W}\|_F^2 - \|\boldsymbol{W}_{\text{High-Energy}}\|_F^2}{\|\boldsymbol{W}\|_F^2} = 1 - P_k.$$

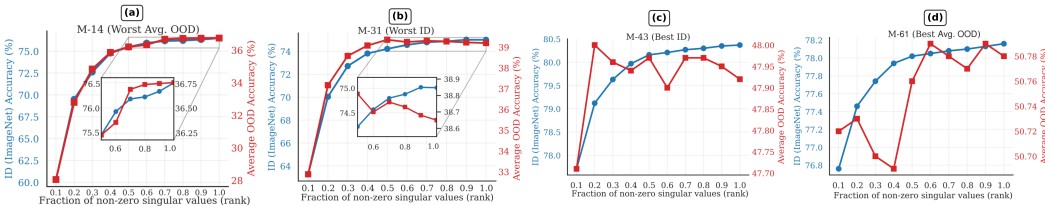

Figure 6: The task vector rank consistently enhances performance on **both** ID and OOD benchmarks. The x-axis represents the rank of the task vector, with blue curves indicating ID accuracy and red curves depicting average OOD accuracy.

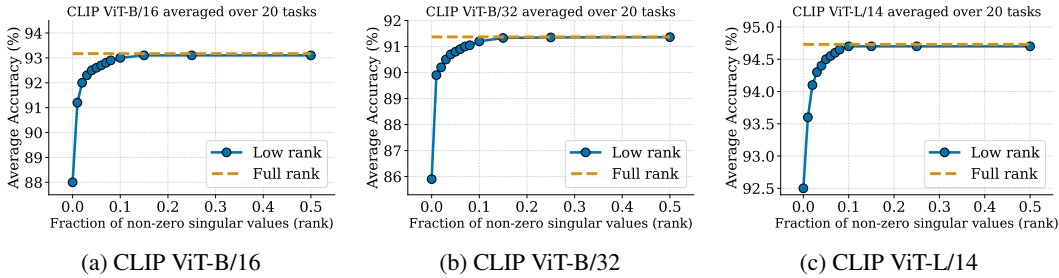

(a) CLIP ViT-B/16        (b) CLIP ViT-B/32        (c) CLIP ViT-L/14

Figure 7: Mean absolute accuracy of the CLIP ViT-{B/32, B/16, L/14} models across increasing fractions of retained singular components, averaged over 20 tasks released by (Wang et al., 2024a). The yellow line represents the average accuracy of the original fine-tuned models with full-rank task matrices, while the blue line shows the accuracies using low-rank approximations.

**Theorem 1.** *The angle parameter $\alpha$ satisfies the following bounds:*

$$\max\left(0, 1 - R - \frac{\sigma_k^2}{\|\boldsymbol{W}\|_F^2}\right) < \cos^2\alpha \leq 1 - R.$$

*Proof.* From the relationship $\cos^2\alpha = 1 - P_k$ and the inequality $R \leq P_k$, we immediately obtain

$$\cos^2\alpha \leq 1 - R.$$

For the lower bound, when $k > 1$, we have $P_k - \sigma_k^2/\|\boldsymbol{W}\|_F^2 < R$, which yields

$$1 - \cos^2\alpha - \frac{\sigma_k^2}{\|\boldsymbol{W}\|_F^2} < R,$$

and therefore

$$\cos^2\alpha > 1 - R - \frac{\sigma_k^2}{\|\boldsymbol{W}\|_F^2}.$$

Since $\cos^2\alpha \geq 0$, the lower bound becomes

$$\cos^2\alpha > \max\left(0, 1 - R - \frac{\sigma_k^2}{\|\boldsymbol{W}\|_F^2}\right).$$

$\square$

## D    LOW-ENERGY DIRECTIONS

In this section, we present additional observations that build upon and extend the analyses discussed in Figure 3.

In Figure 6, we demonstrate that increasing task vector rank consistently enhances performance on both ID and OOD benchmarks. Our analysis reveals a clear positive correlation between retaining

small singular values, *i.e.*, Low-Energy, and improved generalization performance across diverse model configurations, spanning from high-performing ID models to those with poor average OOD performance. Notably, even when preserving 95% of singular values, performance degradation occurs on both ID and OOD tasks, demonstrating that truncation-based approaches fail to enhance generalization and, counterintuitively, that low-energy components contain critical information for robust performance.

This finding contrasts sharply with our observations on smaller-scale downstream tasks. When replicating truncation experiments across *downstream task arithmetic* operations involving the 20 tasks proposed by (Wang et al., 2024a), the task matrices exhibit pronounced low-rank properties, corroborating previous findings that a limited subset of task vectors can accurately represent each layer's functionality (see Figure 7). Remarkably, retaining only 5% of singular components for each task yields mean accuracy comparable to the original fine-tuned models. This suggests that 95% of singular components in each layer matrix can be removed without significant performance degradation on these smaller-scale benchmarks.

## E  QWEN FINE-TUNING

In this section, we detail the fine-tuning procedure for Qwen. We perform *full-parameter fine-tuning* and train with AdamW (Loshchilov & Hutter, 2019), a batch size of 32, a linear-warmup–cosine-decay learning-rate schedule, and run for 2–5 epochs depending on the experiment. We fine-tune and at a fixed context length of 2,048 tokens to avoid train–test context mismatch.

## F  NOTE ON THE USE OF LARGE LANGUAGE MODELS

We used large language models (LLMs) solely to aid with writing and polishing the manuscript. Specifically, LLMs were used for refining grammar, improving clarity, and rephrasing sentences for readability. All research ideas, experiments, and analyses were conducted independently of LLM assistance.

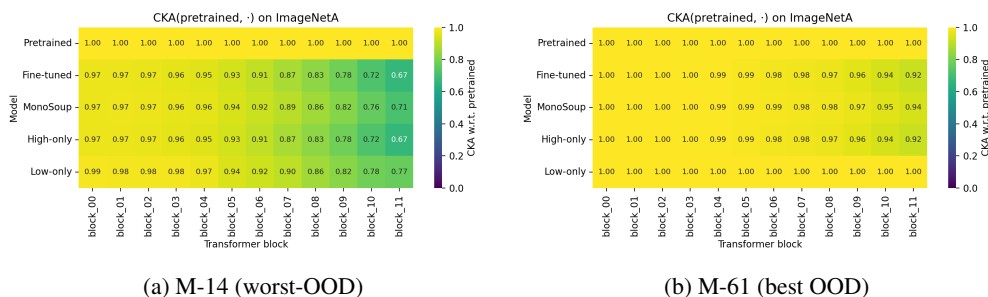

(a) M-14 (worst-OOD)  (b) M-61 (best OOD)

Figure 8: Feature-space alignment maps (CKA). (a) CKA of the model M-14 (worst-OOD) on the ImageNet-A (OOD) dataset (b) CKA of the model M-61 (best-OOD) on the ImageNet-A (OOD) dataset

## G INTERPRETABILITY OF COEFFICIENTS

This section motivates the formulation chosen in Equation 6. We do not claim that $\lambda_{\text{low}}$ is globally optimal under all distributions; rather, we present it as a *simple, principled, and empirically stable* rule derived from clear constraints:

First, we focus on the formula derivation. The coefficient $\lambda_{\text{low}} = \rho + (1 - \rho) \cos \alpha$ follows directly from four natural boundary conditions:

1. suppress low-energy components when the spectrum is sharp and misaligned ($\rho \approx 0$, $\cos \alpha \approx 0$)

2. keep them when the spectrum is flat or strongly aligned ($\rho \approx 1$ or $\cos \alpha \approx 1$)

3. fall back to the spectral baseline when alignment is poor ($f(\rho, 0) = \rho$)

4. rely purely on alignment when low-energy mass is negligible ($f(0, \cos \alpha) = \cos \alpha$).

If we furher assume that the interaction between $\rho$ and $\cos \alpha$ should be bilinear (the simplest smooth form), these four constraints *uniquely determine*

$$f(\rho, \cos \alpha) = \rho + (1 - \rho) \cos \alpha.$$

Thus the formula is not heuristic: it is the *minimal function* satisfying all desired behaviors.

**Convergence and sensitivity.** MonoSoup is a *one-shot* update, so convergence is not applicable. In terms of *sensitivity*, the rule is monotone and 1-Lipschitz in both arguments:

$$\frac{\partial f}{\partial \rho} = 1 - \cos \alpha \in [0, 1], \qquad \frac{\partial f}{\partial c} = 1 - \rho \in [0, 1].$$

Hence small perturbations in $\rho$ or $\cos \alpha$ lead to proportionally small changes in $\lambda_{\text{low}}$.

## H CENTERED KERNEL ALIGNMENT ANALYSIS

For each transformer block $\ell$, we compare the hidden features of: (a) Pre-trained, (b) fine-tuned, (c) MonoSoup, (d) High-only ($W_{\text{High}}$), (e) Low-only ($W_{\text{Low}}$). We compute linear CKA (Kornblith et al., 2019) to the pre-trained features on an unlabeled set from ImageNet-1K and OOD set.

In this ablation analysis, we aim to know if MonoSoup "keeps pre-trained knowledge while retaining specialization." Its features should stay closer to the pre-trained than the fully fine-tuned model on OOD, yet not collapse to pre-trained on ID. Subsequently, we plot the layer-by-layer heatmap of CKA (pretrained, model features) for (a)-(e) on the ID vs. OOD dataset. We show this analysis in Figure 8, for the worst-OOD model and the best-OOD model.

Due to the computational cost of calculating the CKA for each transformer block, we conduct this experiment using 25 batches of size 256 on ImageNet (ID) data and the most challenging OOD dataset among ImageNet distribution shifts, which is ImageNet-A (OOD). This setup ensures that all samples from ImageNet-A are included within the specified number of batches and samples per batch.

We first plot the pretrained row in the figures above as a sanity check, confirming it remains at $1.00$. In the M-14 plot for the "Fine-tuned" row, early blocks (0–3) exhibit CKA values around $0.97$, middle blocks (4–6) decline from approximately $0.96$ to $0.93$, and deeper blocks (7–11) progressively decrease from about $0.87$ to $0.67$. Comparing this to the strong model (best-OOD) shown in the right panel of the figure, the deepest layers diverge significantly more from the pretrained ImageNet-A representation. Notably, Block 11 shifts from $0.67$ CKA in the fine-tuned model to $0.92$ in the strong model. Conversely, the MonoSoup row shows early blocks nearly identical to the fine-tuned model ($0.97$), while deeper blocks consistently exhibit higher CKA values than the fine-tuned model; for instance, Block 9: $0.82$ vs. $0.78$, Block 10: $0.76$ vs. $0.72$, and Block 11: $0.71$ vs. $0.67$.

MonoSoup preserves the shallow and mid layers while noticeably shifting the deepest layers closer to the pretrained ImageNetA representation. This visually demonstrates how MonoSoup partially de-specializes the model, enhancing OOD robustness while retaining most of the fine-tuned signal. The "High-only" row closely mirrors the "Fine-tuned" model, with early blocks showing identical values around $0.97$ and later blocks following the same declining pattern, ending near $0.67$. This clearly illustrates the decomposition's insight: the high-energy task directions ($\Delta \boldsymbol{W}_{\text{high}}$) essentially represent the fine-tuning, such that $\boldsymbol{W}_0 + \Delta \boldsymbol{W}_{\text{high}} \approx \boldsymbol{W}_1$. These directions capture the majority of the representational drift from pretraining, which empirically corresponds to the drift that degrades OOD performance in this weak model (M-14: worst-OOD). This encapsulates the "truncate hurts OOD" phenomenon in feature space, where retaining only $\Delta \boldsymbol{W}_{\text{high}}$ reproduces the OOD-vulnerable representation.

The Low-only model remains closest to the pretrained representation, showing better OOD performance but poorer in-distribution (ID) results. In contrast, the High-only model aligns with the Fine-tuned model and inherits its OOD vulnerabilities. MonoSoup smoothly interpolates between these extremes, balancing the trade-off. This aligns with our hypothesis: Low-only retains old knowledge at the cost of ID specialization, exhibiting the highest CKA to pretrained but worst ID performance; High-only represents pure task directions, diverging most from pretrained in deep layers and suffering worst OOD; MonoSoup combines these, achieving closer alignment to pretrained than Fine-tuned or High-only on OOD without reverting fully to pretrained features.

On the worst-OOD model like M-14, fine-tuning significantly distorts deep features away from the pretrained representation on OOD data (ImageNetA), with high-energy task directions embodying this distortion. MonoSoup mitigates this by reweighting high- and low-energy updates, notably pulling the deepest layers closer to the pretrained representation (CKA increase of $0.04$–$0.05$), which corresponds to the observed OOD performance gains. Conversely, on stronger backbones where fine-tuning already maintains a high similarity to pretrained features (CKA $\geq 0.9$), MonoSoup's adjustments and accuracy improvements are naturally smaller. This exemplifies the principle that MonoSoup is most effective when fine-tuning begins to disrupt the pretrained representation, as demonstrated by two models and their corresponding CKA heatmaps.

Recall that $\boldsymbol{W}_{\text{Low}}^{(\ell)}$ is the component whose magnitude is modulated by $\cos \alpha^{(\ell)}$ through $\lambda_{\text{low}}^{(\ell)}$ in Equation 6. The fact that the Low-only model remains closest to the pre-trained representation (highest CKA) and improves OOD at the cost of ID, while High-only mirrors the fine-tuned representation and inherits its OOD vulnerabilities, empirically validates our interpretation of $\cos \alpha^{(\ell)}$ as a *pre-training preservation signal*: increasing $\lambda_{\text{low}}^{(\ell)}$ (hence $\cos \alpha^{(\ell)}$) shifts the representation toward the pre-trained solution on OOD inputs in exactly the way predicted by our CKA analysis.

# I   VARIANCE THRESHOLD ($R$) ABLATION

In this section, our aim is to analyze the role of the hyperparameter $R$ as our variance threshold.

**Zero-Shot initialization.**   To assess the generalizability of our method, we evaluate its performance on two CLIP ViT-B/32 models that were fine-tuned from zero-shot initialization (models

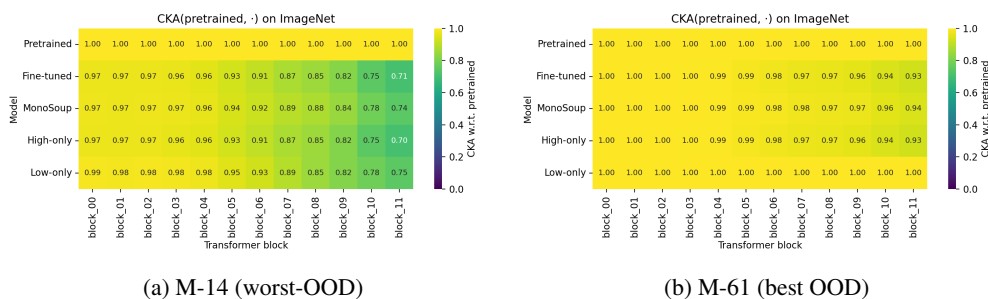

(a) M-14 (worst-OOD)        (b) M-61 (best OOD)

Figure 9: Feature-space alignment maps (CKA). (a) CKA of the model M-14 (worst-OOD) on the ImageNet (ID) dataset (b) CKA of the model M-61 (best-OOD) on the ImageNet (ID) dataset

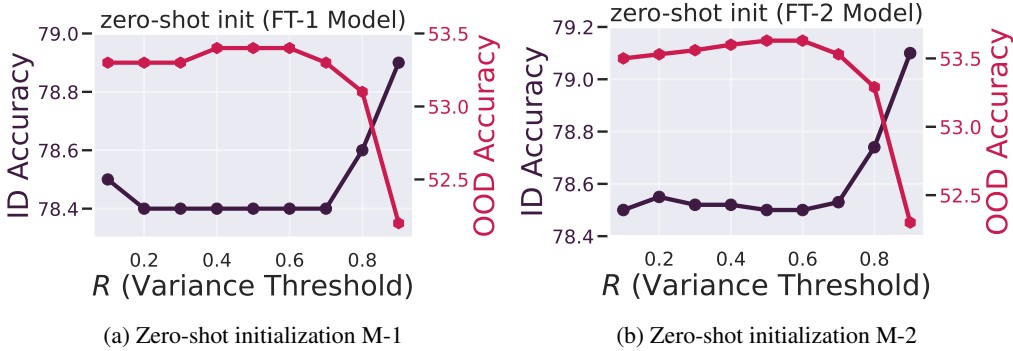

(a) Zero-shot initialization M-1        (b) Zero-shot initialization M-2

Figure 10: Ablation on variance threshold $R$.

utilized in Appendix B), rather than from linear probing initialization. Figure 10 presents this analysis.

**ConvNeXt model.** In this part, we assess the generalizability of our method on ConvNeXt[1] (Liu et al., 2022) model pretrained on ImageNet-22k and fine-tuned on ImageNet-1k (See Figure 11).

**CLIP (FT-1, zero-shot init) Figure 10.** Varying the variance threshold $R$ reveals a clear trade-off between ID and OOD performance. OOD accuracy rises from $\approx 53.1\%$ at small $R$ to a plateau/peak around $53.3 - 53.4\%$ for $R \in [0.4, 0.6]$, then drifts down as $R \to 0.9$. ID accuracy is essentially flat near $78.4\%$ for $R \leq 0.7$ and only ticks up at high thresholds (to $\approx 78.9\%$ at $R = 0.9$). Interpreting $R$ as "how many high-energy directions we keep," this says moderate truncation preserves low-energy residuals that help OOD, while very large $R$ steers the model closer to the fine-tuned solution and benefits ID at the expense of OOD. A pragmatic operating point is $R \approx 0.5 - 0.7$ if OOD is prioritized, and $R \geq 0.85$ if ID is.

**CLIP (FT-2, zero-shot init) Figure 10.** The second CLIP model shows the same shape, confirming the effect is not idiosyncratic. OOD accuracy improves slightly as $R$ increases from 0.1 and peaks near $R \approx 0.5$ (low-53% range), then declines toward $\approx 52.5\%$ at $R = 0.9$. ID accuracy exhibits a shallow U-shape: small changes for $R \leq 0.7$, then a marked increase at $R \in [0.8, 0.9]$ (to $\sim 79.1\%$). Again, mid-range $R$ balances the objectives, while pushing $R$ high recovers ID at clear OOD cost. The near-identical trends across the two fine-tuned CLIP checkpoints strengthen the interpretation that $R$ primarily trades low-energy residual cues (good for OOD) against a higher-rank approximation of the fine-tuned update (good for ID).

**ConvNeXt Figure 11.** Sensitivity is gentler and largely monotonic: as $R$ increases from 0.1 to 0.9, ImageNet accuracy climbs from $\approx 84.9\%$ to $\approx 85.6\%$ and average OOD accuracy rises from

---

[1]checkpoint's link

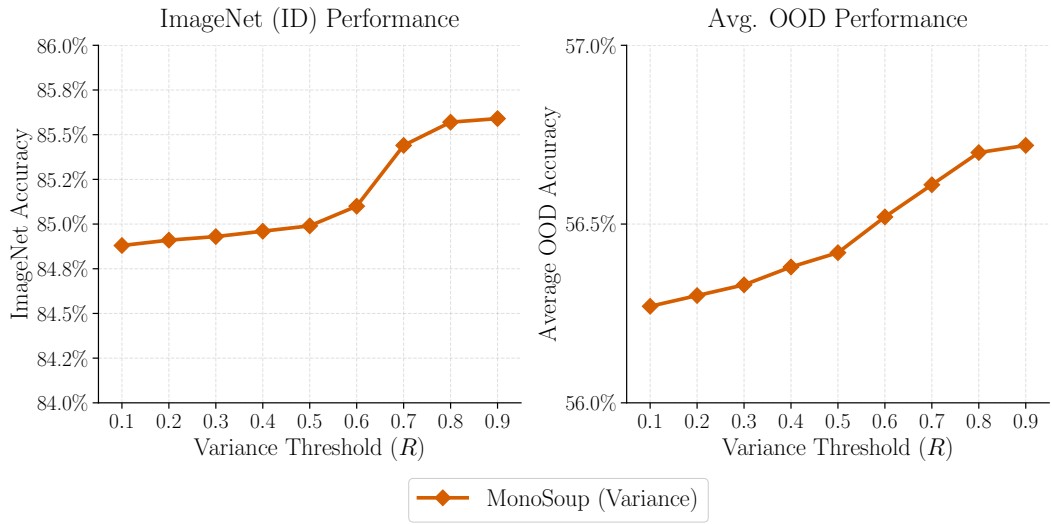

Figure 11: Ablation study on the variance threshold $R$ according to the ConvNeXt model.

$\approx 56.2\%$ to $\approx 56.9\%$. Unlike CLIP, ConvNeXt shows little ID-OOD tension; both benefit as more high-energy directions are retained, with a modest knee around $R \approx 0.7 - 0.8$. This suggests the ConvNeXt fine-tuning delta is spectrally more "benign": expanding the kept subspace steadily recovers useful signal without discarding low-energy components that disproportionately help OOD. As a default, $R \in [0.7, 0.85]$ delivers the best joint ID-OOD trade-off across all three studies.

## J  CONVNEXT

In this section, we assess the generalizability of our method on other architectures rather than CLIP models with ConvNeXt[2] (Liu et al., 2022) represents performance of the MonoSoup on these models.

| Method | Performance Metrics | |
|---|---|---|
| | In-distribution (ImageNet) | Avg OOD |
| *ConvNeXt* | | |
| Pretrained (IN12k) | 82.17 % | 52.88 % |
| Fine-tuned (IN1k) | 85.17 % | 55.85 % |
| MonoSoup (freevariance) | 85.19 % | 56.07 % |
| MonoSoup (variance) $R = 0.8$ | **85.57 %** | **56.70 %** |

Table 5: Performance of ConvNeXt model.

---

[2]checkpoint's link

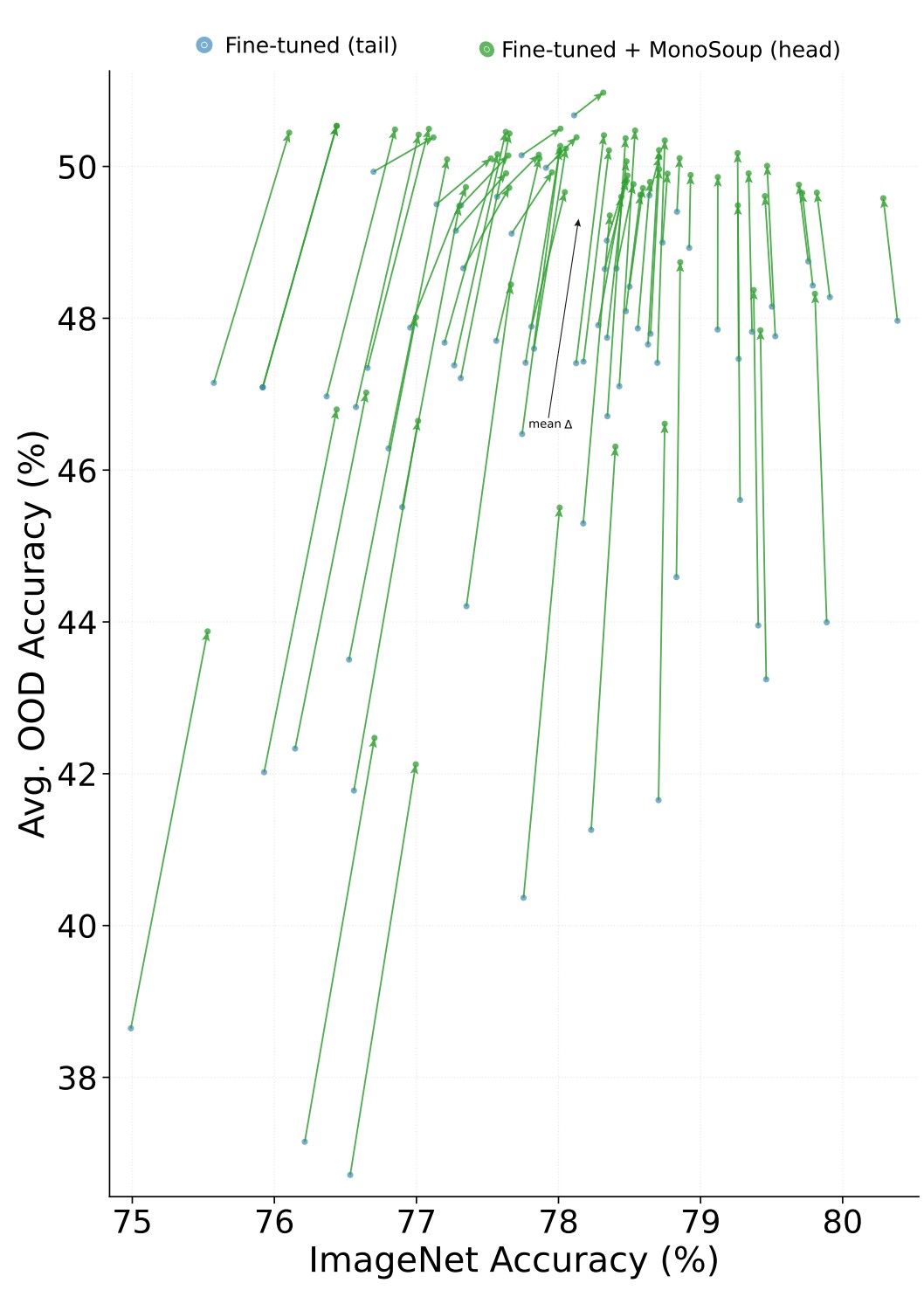

Figure 12: **Quiver Plot.** To demonstrate robustness across a variety of models, we further assess our method using all 70 CLIP checkpoints provided by (Wortsman et al., 2022a). We present a vector plot in which the x- and y-axes represent ID and OOD performance, respectively. Each vector begins at the performance level of a fine-tuned checkpoint and extends to the performance achieved after the application of MonoSoup.

