# OpenReview forum: "Model soups need only one ingredient"
_ICLR.cc/2026/Conference — Submitted to ICLR 2026_

### Official Review · Reviewer_Yqvj · 2025-10-24

**Soundness:** 3
**Presentation:** 3
**Contribution:** 3
**Rating:** 6
**Confidence:** 4

**Summary:**

In the paper the authors introduce MonoSoup, a datafree approach that tries to obtain the same benefits of Model soups (recover ood perfomenace after finetuning) while using only a single checkpoint instead of averaging multiple checkpoints. The method applies Singular Value Decomposition (SVD) to each layer update in order to split into high and low energy directions. Monosoup then re-weights these directions with adaptive and layer-wise coeffiecients computed taking into account the spectral and geometric structure of the model. The authors provide experiments on both visual and text domain, using CLIP as reference model for vision domain and Qwen as model for text.

**Strengths:**

Simple and straightforward method: easy to implement and interpret.

Well-motivated and theoretically rooted in spectral theory: the work builds on the spectral and geometric understanding of model merging, offering clear insights into when and why merging succeeds.

Strong analysis and motivation: the introduction of Similarity-Filtered Greedy Soup empirically validates that spectral alignment is a strong proxy for effective merging, motivating the proposed method.

Efficiency: achieves the benefits of model soups while requiring only a single checkpoint, MonoSoup performs on par with or better than Model Soup and Model Stock while being much cheaper to apply.

Complementary with existing techniques: for example combining the method with Wise-FT further improves the ID–OOD performances.

**Weaknesses:**

Modest gains on strong checkpoints: While MonoSoup provides clear improvements on weaker or representation-collapsed models, it yields only modest gains on already well-performing checkpoints. In those cases, the improvements over Model Soup or Model Stock are often within the margin of variability, suggesting that the method’s primary advantage lies more in efficiency rather than in delivering substantial performance breakthroughs for strong models.

Lack of variability reporting: The experimental tables report mean accuracies but omit measures of variability such as standard deviations. Given that several of the reported improvements are relatively small (often around 0.5–1%), including standard deviations would help assess the statistical significance and robustness of these gains. The absence of such information makes it difficult to judge whether improvements are consistent or within the expected variance of fine-tuning noise.

Limited interpretability of coefficients: while layer-wise coefficients are derived analytically, their per-layer impact on robustness is not deeply analyzed.

**Questions:**

For the models labeled M-x (e.g., M-14, M-31, etc.), could the authors clarify what these identifiers refer to? Do they correspond to specific fine-tuning configurations ?

---

> ### Author Response · Authors · 2025-11-25
>
> We thank Reviewer Yqvj for their thoughtful and constructive feedback. We appreciate that they find MonoSoup to be straightforward, well-motivated, theoretically grounded, and efficient. Below we address the individual points and clarify several aspects of the work.
>
> > W1: Modest gains on strong checkpoints
>
> We appreciate the reviewer’s observation that improvements on already well-performing checkpoints can be modest. This is an inherent consequence of the problem setting: MonoSoup operates on a single, fixed, fine-tuned checkpoint, without training, data, or multiple models to leverage. Therefore, gains are limited to the underlying checkpoint they operate on. Because strong checkpoints are already highly optimized, they contain little harmful noise and limited representational collapse—hence any post-hoc editing method can only make incremental improvements. For instance, in Table 1 of the original Model Soups paper (which even leverages multiple checkpoints), the best ViT-G model from hyperparameter search achieves 90.78 and 84.68, respectively, while Greedy Soup achieves 90.94 and 85.02.
>
> That said, MonoSoup provides consistent ID and OOD gains across all checkpoints, and substantial improvements on weaker or collapsed ones (e.g., +7–8% OOD in Table 1). Importantly, the method does not degrade a model, as it refines its update anisotropically rather than replacing or averaging weights. MonoSoup is inherently complementary to existing merging approaches, e.g., Soups or Model Stock, as it operates on a single checkpoint. This is particularly important beyond well studied academic benchmarks such as the one proposed by the original Model Soups paper.
>
> > W2: On the lack of variability reporting
>
> Our tables do not include variability measures because we report results for specific fixed checkpoints, and the MonoSoup edit is entirely deterministic—there is no source of stochasticity once the checkpoint is chosen. To better illustrate robustness across a broad range of models, we additionally evaluate our method on all 70 checkpoints released by the original Model Soups paper. In the rebuttal, we include a vector plot in Figure 12 where the x- and y-axes correspond to ID and OOD performance, respectively, with each vector originating at the performance of a fine-tuned checkpoint and pointing to the performance after applying MonoSoup. We observe that MonoSoup results in significant gains across this collection of models, especially for OOD performance.
>
> > W3: Limited interpretability of coefficients
>
> We thank the reviewer for their thoughtful question. Our goal is not to claim that $\lambda_{\text{low}}$ is globally optimal under all distributions, but rather that it is a *simple, principled, and empirically stable* rule derived from clear constraints. We will clarify this in the revision.
>
> First, we focus on the formula derivation. The coefficient $\lambda_{\text{low}} = \rho + (1 - \rho)\cos \alpha$ follows directly from four natural boundary conditions:
> - suppress low-energy components when spectrum is sharp and misaligned ($\rho \approx 0$, $\cos \alpha \approx 0$);
> - keep them when the spectrum is flat or strongly aligned ($\rho \approx 1$ or $\cos \alpha \approx 1$);
> - fall back to the spectral baseline when alignment is poor ($f(\rho, 0) = \rho$);
> - rely purely on alignment when low-energy mass is negligible ($f(0, \cos\alpha) = \cos\alpha$).
>
> If we furher assume that the interaction between $\rho$ and $\cos\alpha$ should be bilinear (the simplest smooth form), these four constraints *uniquely determine*
>
> $$
> f(\rho, \cos\alpha) = \rho + (1 - \rho)\cos\alpha .
> $$
>
> Thus the formula is not heuristic: it is the **minimal function** satisfying all desired behaviors.
>
> Regarding the reviewer's concerns about convergence and sensitivity, MonoSoup is a one-shot update, so convergence is not applicable. In terms of **sensitivity**, the rule is monotone and $1$-Lipschitz in both arguments:
>
> $$
> \frac{\partial f}{\partial \rho} = 1 - \cos\alpha \in [0, 1],
> \qquad
> \frac{\partial f}{\partial c} = 1 - \rho \in [0, 1].
> $$
>
> Hence small perturbations in $\rho$ or $\cos\alpha$ lead to proportionally small changes in
> $\lambda_{\text{low}}$.
>
> We hope this clarifies the motivation and structure behind the coefficient design and addresses the reviewer's concerns.
>
>
> > Q: Clarifying the meaning of the M-x model identifiers
>
> Yes, these are indexes of 70 models released by the original model soups paper. M-0 refers to initialization, while 1 to 71 indicates the index of each fine-tuned model, which has been fine-tuned using different hyperparameters. These details can be found in a JSON file within their [GitHub repository](https://github.com/mlfoundations/model-soups/blob/main/hparam_info.json)

---

> > ### Comment · Reviewer_Yqvj · 2025-11-26
> >
> > Thank you for the clarifications. The explanations regarding the limited gains on strong checkpoints and the principled derivation of the layer-wise coefficients are helpful. The interpretation of the M-x identifiers is now clear as well. I appreciate the additional analysis across the full set of 70 models, which further contextualizes the behavior of MonoSoup.
> > I will adjust my score accordingly.

---

> > > ### Author Response · Authors · 2025-11-26
> > >
> > > We thank the reviewer for their follow-up and for acknowledging the value of our clarifications. We are pleased that the explanations regarding the layer-wise coefficients and the behavior on strong checkpoints effectively addressed your concerns.
> > > We also appreciate the positive reception of the additional analysis across the full set of 70 models. We agree that this large-scale validation provides crucial context for MonoSoup, and we thank you for prompting this improvement. We are grateful for your reassessment and the decision to adjust the score.

---

### Official Review · Reviewer_gzvH · 2025-10-28

**Soundness:** 3
**Presentation:** 3
**Contribution:** 3
**Rating:** 6
**Confidence:** 5

**Summary:**

This paper proposes MonoSoup, a data-free, single-checkpoint approach that recovers the robustness benefits of model soups without needing multiple models. For each layer, the fine-tuning update is decomposed into high- and low-energy components (via SVD), and adaptive, per-layer interpolation coefficients are computed to recombine them. This reweighting preserves task-specific signal while retaining useful low-energy structure, leading to improvements in both in-distribution accuracy and out-of-distribution robustness. Results on vision and language tasks show consistent gains over strong baselines.

**Strengths:**

# Originality

The contribution is not a new mathematical primitive but a **creative reframing** of model-soup robustness: the authors show how to extract and reweight **within-checkpoint** structure so that a *single* fine-tuned model can capture much of the benefit previously obtained by averaging *multiple* checkpoints. This shift from multi-model soups to a **single-checkpoint soup** removes the dependency on collecting and storing many models while preserving the core geometric insight behind soups.

# Quality

The work is **methodologically sound**: the approach is clearly specified; comparisons include strong, relevant baselines; and the analysis (including ablations) is thoughtful and aligns with the stated hypotheses. Empirical results are broad enough to support the main claims and are reported with appropriate care.

# Clarity

The paper is **well written and well structured**. The narrative flows logically from diagnosis to method to evaluation, with enough technical detail to enable reproduction.

# Strengths

* **Conceptual creativity:** Derives a soup-like effect from a **single checkpoint** via principled layer-wise decomposition and reweighting.
* **Practical impact:** **Eliminates the burden** of curating and storing many checkpoints, reducing compute and storage overhead.
* **Empirical thoroughness:** **Extensive evaluations** with clear, interpretable metrics and ablations that illuminate why the method works.
* **Presentation quality:** Clear exposition, intuitive figures, and actionable implementation details.

**Weaknesses:**

* **“Plug-and-play” claim needs qualification.** The method’s practicality is somewhat overstated because performance depends on choosing (R) well. Without a label-free selection rule, users may not reliably reproduce the reported gains.

* **Limited architecture diversity.** Most results are on Transformer backbones (CLIP ViTs, a small LLM). It remains unclear whether the recommended (R) range and mixing rule transfer to non-Transformer models (e.g., ConvNeXt, ResNet, MoE).

* **Architecture- and dataset-dependence of (R).** Intuitively, the optimal (R) should vary with model family and pretraining corpus, yet the paper suggests a fairly stable mid-range (R) for ViTs across datasets without explaining why.

**Questions:**

see weakness,

**Hyperparameter selection for (R).**
While the paper argues that performance is fairly stable for mid-range (R), the work would benefit from a **practical, label-free procedure** to set (R). In particular, because spectral decay and alignment statistics can vary across architectures (e.g., ViT vs. ConvNets vs. LLMs) and datasets, a fixed default may not transfer. Please (i) justify when/why a mid-range (R) should generalize, and (ii) provide a simple selection rule that practitioners can use without labels.

---

> ### Author Response · Authors · 2025-11-25
>
> We thank Reviewer gZvH for their thoughtful and positive evaluation. We appreciate the recognition of the method’s conceptual reframing, methodological clarity, and empirical strengths. Below we address the reviewer’s concerns regarding the selection of the variance threshold \(R\), which appear across multiple weaknesses and in the questions section.
>
> ## Effective-Rank Rule (ERank-MonoSoup)
>
> We thank the reviewer for repeatedly emphasizing the importance of making MonoSoup fully plug-and-play and less sensitive to the choice of the variance threshold \(R\). Following [Roy & Vetterli (2007)](https://www.eurasip.org/Proceedings/Eusipco/Eusipco2007/Papers/a5p-h05.pdf), we introduce **ERank-MonoSoup**, an adaptive mechanism based on the **effective rank** of each layer’s singular-value spectrum. Specifically, the ERank variant performs the following steps:
> Normalize singular values:  $p_i = \frac{\sigma_i}{\sum_j \sigma_j}$
> Calculate the entropy $H = -\sum_i p_i \log p_i$
> The effective rank is  $r_{\mathrm{eff}} = \exp(H)$
> And we turn this into an integer by setting $(k = \lceil r_{\mathrm{eff}} \rceil)$, which induces $R = P_k$, the cumulative spectral mass up to index $k$.
>
> Equations (4)–(6) of the method then proceed unchanged.  This rule is **fully label-free**, **automatically adapts** to each layer’s spectral shape, and is **model-agnostic**. We will include ERank-MonoSoup as an additional variant in the final manuscript.
>
> We now proceed to answer each concern below.
>
> > W1: Plug-and-play” claim needs qualification.
>
> By “plug‑and‑play” we mean that MonoSoup is data‑free, post‑hoc and edits a  single‑checkpoint with a single, interpretable hyperparameter $R$. Along with the new variant introduced above, we would like to ask the reviewer to reconsider their score.
>
> > W2 & W3: Limited architecture diversity.
>
> We thank the reviewer for their suggestion. We expand the experiments on ConvNext architectures, where we see that MonoSoup variants (fixed variance and the proposed ERank) lead to improvements.
>
> | Method | ID Acc | Avg. OOD |
> |--------|--------|-----------|
> | ConvNeXt Pretrained (IN12k) | 0.8217 | 0.5288 |
> | ConvNeXt FT (IN1k) | 0.8517 | 0.5585 |
> | MonoSoup (ERank) | 0.8598 | 0.5618 |
> | MonoSoup (R = 0.8) | 0.8557 | 0.5670 |
>
> Moreover, we include the results of the new variant on the CLIP ImageNet benchmark
>
> | Method | ID Acc (%) | Avg. OOD (%) |
> |--------|------------|---------------|
> | Initialization | 75.4 | 46.2 |
> | FT (OOD+) | 78.11 | 50.67 |
> | + MonoSoup (R = 0.8) | 78.29 | 51.60 |
> | + ERank-MonoSoup | 78.21 | 50.91 |
> | FT (OOD-) | 76.53 | 36.71 |
> | + MonoSoup (R = 0.8) | 78.55 | 44.21 |
> | + ERank-MonoSoup | 78.03 | 42.78 |
> | FT (ID+) | 80.38 | 47.96 |
> | + MonoSoup (R = 0.8) | 80.03 | 49.95 |
> | + ERank-MonoSoup | 80.34 | 48.94 |
> | FT (ID-) | 74.99 | 38.64 |
> | + MonoSoup (R = 0.8) | 77.76 | 46.54 |
> | + ERank-MonoSoup | 77.38 | 45.22 |
>
>
>
> > Q: Hyperparameter selection for R
>
> Apart from the above practical and label-free procedure, as per the reviewer’s suggestion, we have also included in the updated manuscript ablations on R in Figures 10 and 11 of the appendix.
>
> CLIP (FT-1, zero-shot init). OOD peaks for $R \in[0.4,0.6](\sim 53.3-53.4 \%)$ and degrades as $R \rightarrow 0.9$; ID is flat near $78.4 \%$ until $R \geq 0.8$ where it rises to $\sim 78.9 \%$. In short: mid- $R$ favors OOD, very high $R$ favors ID. Use $R \approx 0.5-0.7$ when OOD matters, $R \geq 0.85$ when ID does.
>
> CLIP (FT-2, zero-shot init). Same shape: OOD improves then peaks near $R \approx 0.5$ and drops to $\sim 52.5 \%$ at $R=0.9$; ID shows a mild U-shape with a lift at $R \in[0.8,0.9](\sim 79.1 \%)$. This replicates the trade-off: moderate truncation preserves low-energy cues useful for OOD; very high $R$ tracks the fine-tuned solution and helps ID.
>
> ConvNeXt. Sensitivity is gentler and mostly monotone: as $R$ increases $0.1 \rightarrow 0.9$, ID rises $\sim 84.9 \rightarrow \sim 85.6 \%$ and avg-OOD $\sim 56.2 \rightarrow \sim 56.9 \%$, with a soft knee around $R \approx 0.7-0.8$. Unlike CLIP, there's little ID/OOD tension; $R \in[0.7,0.85]$ is a robust default.
>
> Further diagnostics-including layer-wise CKA maps showing how edited models relate to the pre-trained and fine-tuned representations-are in the appendix.

---

> > ### Comment · Reviewer_gzvH · 2025-11-26
> >
> > I thank the authors for their detailed response. The introduction of the ERank-MonoSoup method effectively addresses my core concerns regarding hyperparameter selection and the 'plug-and-play' claim. While architectural diversity has only been partially addressed, the additional ConvNeXt experiments provide sufficient evidence of transferability. Consequently, I am inclined to increase my score.

---

> > > ### Author Response · Authors · 2025-11-26
> > >
> > > We sincerely thank the reviewer for their follow-up and for acknowledging that our additional experiments and the introduction of the ERank-MonoSoup method have addressed your core concerns. We appreciate your constructive feedback throughout this process, which has significantly improved the robustness and clarity of our work.
> > > We are grateful for your reassessment and your decision to increase the score.

---

### Official Review · Reviewer_q1EA · 2025-10-29

**Soundness:** 2
**Presentation:** 3
**Contribution:** 2
**Rating:** 4
**Confidence:** 5

**Summary:**

This paper addresses a key limitation of existing model merging methodologies, pointing out their reliance on two or more checkpoints. The authors argue this is misaligned with realistic scenarios where often only a single, best-performing model is stored. To address this gap, the paper proposes MonoSoup, a method that utilizes only a single checkpoint.
Using SVD, the method decomposes the single model's update into a high-energy component, which is associated with in-distribution (ID) specialization, and a low-energy component for out-of-distribution (OOD) robustness. By performing a principled weighted sum of these two components, MonoSoup demonstrates performance that is comparable to existing multi-checkpoint techniques.

**Strengths:**

1. The paper addresses a practical and realistic scenario. While many merging methods like Model Soups require access to dozens of checkpoints, real-world applications often only store a single, best-performing model. The paper's focus on improving robustness from this single-mode" setting is a valuable contribution.
2. The core idea of using SVD to decompose the fine-tuning update into high-energy (specialization) and low-energy (robustness) components is interesting.

**Weaknesses:**

1. The paper's method for calculating the alignment coefficient $cos~\alpha^{(l)}$ (lines 264-269) is theoretically unclear and appears to be a significant conceptual leap. The method computes the cosine similarity between two conceptually different entities: the pre-trained weights $W_0^l$ (an absolute state vector) and the low-energy update $W_{low}^l$ (a difference vector). There is no clear justification for why the alignment between a 'state' and a 'difference' is a meaningful measure of knowledge preservation. Even if $W_0^l$ is interpreted as a vector from the origin, it is not the pre-training update direction (unless initialized from zero), making the interpretation of this alignment ambiguous.
2. The paper's key motivation is undermined by its own data in Figure 3b. The text claims that truncating low-energy components harms both ID and OOD accuracy, motivating the need for re-weighting (MonoSoup). However, the graph clearly shows that moderate truncation (e.g., at a rank fraction of $\approx 0.7$) improves OOD performance (red line).
3. The performance gains over LiNeS reported in Table 2 are marginal. These small improvements might not be statistically significant and could be due to experimental variance or checkpoint selection. The claim of consistent improvement should be validated more robustly, for instance, by reporting results over multiple runs or across a wider set of fine-tuned checkpoints (beyond the three presented).
4. The method's generality appears limited. The key motivation (low-energy components are crucial for robustness) is derived only from the large-scale ImageNet benchmark (Figure 3b) and is explicitly shown to be absent in the small-scale 20-task benchmark. This suggests MonoSoup may only be effective for fine-tuning on large-scale datasets. This contradiction also weakens the general claim that low-energy components inherently encode OOD robustness, as this does not hold for the small-task setting.

**Questions:**

1. The paper argues that high task vector alignment is beneficial for merging models performing the same task. This contrasts with principles in related fields like multi-task learning, where orthogonality (low similarity) is often preferred to prevent task interference[1,2]. Could the authors clarify how to reconcile these two perspectives? Is the principle that "high alignment is beneficial" strictly limited to the same-task merging scenario?
2. In Figure 2, the authors state they analyze 2,409 pairwise combinations. However, $70 \choose 2$ (choosing 2 pairs from 70 models) should result in 2,415 combinations. Could the authors clarify this small discrepancy?
3. The symbol $\tau$ is used in the context of Similarity-Filtered Greedy Soup without a formal definition.
4. Table 1 introduces "MonoSoup (Pairwise Models)" as a baseline. The main text, however, focuses on applying MonoSoup to a single model. Could the authors briefly explain in the main text how MonoSoup is adapted and applied in the pairwise setting?

[1] Ilharco, Gabriel, et al. "Editing models with task arithmetic." ICLR (2023).
[2] Davari, MohammadReza, and Eugene Belilovsky. "Model breadcrumbs: Scaling multi-task model merging with sparse masks." ECCV (2024).

---

> ### Author Response · Authors · 2025-11-25
>
> > W1: Clarification on the motivation and interpretation of the alignment coefficient.
>
> In our formulation, high-energy directions capture the main task-specific adaptation (specialization), while low-energy directions represent residual updates. These residuals can be weak specialization, noise, or sometimes robustness-relevant structure. The purpose of the alignment term $\cos\alpha^{(\ell)}$ is not to decide whether the low-energy component is “signal vs. noise,” but rather to test whether these residual directions reuse features already present in the pre-trained weights.
>
> - When $\cos\alpha^{(\ell)} \approx 1$, the low-energy subspace strongly overlaps with $W_0^{(\ell)}$. We interpret this as refining or rescaling existing pre-training features, and we keep this component with a larger weight (higher $\lambda^{(\ell)}_{\text{Low}}$).
> - When $\cos\alpha^{(\ell)} \approx 0$, the low-energy subspace is nearly orthogonal to $W_0^{(\ell)}$. In this case, we have little evidence that these residual directions correspond to preserved pre-training knowledge, so we treat them as likely noise or overly idiosyncratic specialization and down-weight them. Importantly, the main task-specific signal is already captured in the high-energy component $\Delta W^{(\ell)}_{\text{High}}$; $\cos\alpha^{(\ell)}$ only controls how cautiously we retain the residual low-energy part.
>
> Combined with the spectral term $\rho^{(\ell)}$, low-energy directions are strongly suppressed only when both their spectral mass and their alignment with $W_0^{(\ell)}$ are small. Empirically, this regime corresponds to updates whose removal improves robustness without harming in-distribution accuracy.
>
>
>
> > W2:  Relationship between truncation results (Fig. 3b) and the motivation for reweighting
>
> To clarify the role of low-energy components and address the reviewer’s concern regarding the motivation in Figure 3b, we conducted a feature-space analysis using linear CKA across all transformer blocks. This allows us to directly measure how fine-tuning, and each component of our decomposition, affect the representation on ID and OOD data. The details are provided in the new Appendix G.
>
> Experimental Setup. For each transformer block $\ell$, we compare the hidden features of: (a) Pre-trained, (b) Fine-tuned, (c) MonoSoup, (d) High-only $\left(W_{\text {High }}\right)$, (e) Low-only $\left(W_{\text {Low }}\right)$. We compute linear CKA to the pre-trained features on an unlabeled set from ImageNet-1K and OOD set.  Due to the computational expense of calculating the CKA for each transformer block, we conduct this experiment using 25 batches of size 256 on ImageNet (ID) data and the most challenging OOD dataset among ImageNet distribution shifts, which is ImageNet-A (OOD). This setup ensures that all samples from ImageNet-A are included within the specified number of batches and samples per batch. In the appendix, we provide layer-by-layer heatmaps of CKA (pretrained, model features) for (a)-(e) on the ID vs. OOD dataset.
>
> We present below our key findings, trying to answer the question if MonoSoup "keeps pre-trained knowledge while retaining specialization."
> Fine-tuning distorts deep representations, especially in weak models. For the worst-OOD checkpoint (M-14), deep blocks diverge markedly from the pretrained features on ImageNet-A (CKA drops from ~0.97 in early blocks to 0.67 in Block 11). This drift is strongly correlated with OOD degradation. In contrast, the strong model (M-61) preserves high similarity to pretrained features across all blocks (CKA > 0.92).
> High-only recovers the fine-tuned drift; Low-only recovers pretrained features. The “High-only’’ model nearly matches the fine-tuned representation, including its OOD-vulnerable deep-layer drift. Conversely, “Low-only’’ stays closest to pretrained and achieves the best OOD but weak ID performance. This directly confirms the decomposition interpretation:
> $ΔW_{high}$ = specialization drift,
> $ΔW_{low}$ = robustness-preserving component.
> MonoSoup selectively corrects deep layers while preserving specialization. MonoSoup keeps shallow and mid-level blocks identical to fine-tuning (CKA ~0.97), but consistently moves deep layers closer to the pretrained representation (e.g., Block 11: 0.67 → 0.71). This matches the observed OOD gains and shows that MonoSoup performs a targeted correction, not a rollback. It reduces harmful drift only where it matters.
>
> We hope that this analysis addresses the concerns regarding Figure 3b: moderate truncation may sometimes improve OOD, but full truncation discards the robustness-preserving low-energy structure. MonoSoup’s reweighting produces stable improvements across checkpoints. Crucially, this also explains why gains are large for weak models (where distortion is severe) and small for strong models (where fine-tuning already preserves pretrained geometry). MonoSoup is most effective precisely when fine-tuning begins to overwrite robust pretrained features.

---

> > ### Author Response · Authors · 2025-11-25
> >
> > > W3: Comparison with LiNeS in Table 2
> >
> > We thank the reviewer for raising the comparison to LiNeS. While both methods improve robustness after fine-tuning, they operate in different regimes. LiNeS requires access to data, as well as two hyperparameters that control the slope and the intercept of the linear mapping. In contrast, MonoSoup is data-free, uses a single, interpretable hyperparameter R, and—as our ablations show—is robust to a wide range of R values. Following the suggestion of Reviewer gZvH, we additionally introduced ERank-MonoSoup, a completely hyperparameter-free variant based on the effective rank of the fine-tuning update. Therefore, MonoSoup is a computationally more viable approach compared to LiNeS that also does not require access to data.
> >
> > > W4: Generality beyond large scale benchmarks
> >
> >
> > Thank you for raising this point. We will clarify why the effect is stronger on ImageNet and why this does not contradict the motivation.
> >
> > First, large-scale fine-tuning induces larger and higher-rank updates. On ImageNet, fine-tuned checkpoints move substantially in parameter space, and—as we show in our new CKA analysis—deep-layer representations drift far from the pretrained features. This drift directly correlates with OOD degradation, making the separation between high- and low-energy components meaningful and enabling MonoSoup to provide sizable gains.
> >
> > Second, the 20-task benchmark produces much smaller, lower-rank updates that remain close to the pretrained model. As a result, there is far less harmful drift to correct, and improvements are necessarily smaller. This does *not* indicate limited generality; it simply reflects that the baseline fine-tuned models already preserve pretrained robustness. This aligns with our answer to Q1: gains depend on the magnitude of representational drift induced by fine-tuning.
> >
> > We hope this clarifies the point about generality and kindly ask the reviewer to reconsider their score.
> >
> >
> >
> > > Q1: Clarifying how alignment principles relate across single-task and multi-task merging contexts
> >
> > The two settings operate under fundamentally different geometric regimes, which explains the apparent contradiction. In multi-task learning, task vectors correspond to functionally different objectives, and orthogonality between them is beneficial because it reduces task interference and prevents one task from overwriting another. Diversity across task directions is inherent and cannot be avoided.
> > In contrast, same-task fine-tuned models lie in a highly aligned, low-dimensional subspace. Many works—including Stochastic Weight Averaging (SWA) [1]—show that checkpoints trained on the same task differ only by small optimization noise, and averaging (or merging) these aligned directions improves performance because it reinforces the shared “core” solution. In this regime, high alignment is not only expected but desirable, as it amplifies the common signal rather than causing interference.
> > Therefore, “alignment is beneficial’’ is specific to merging models trained on the same task, where the underlying objectives coincide and the useful directions naturally align. Multi-task settings optimize different objectives and require orthogonality to reduce interference. We hope that this has clarified the contradiction.
> > [1] Izmailov, Pavel, et al. "Averaging weights leads to wider optima and better generalization." https://arxiv.org/pdf/1803.05407
> >
> >
> > > Q2: Clarifying the missing 6 Model Stock pairings.
> >
> > Thank you for drawing our attention to this. We originally included all combinations but we decided to exclude 6 pairs because ModelStock shows severe performance degradation on both ID and OOD benchmarks and their inclusion would make the scatter points of Fig 2a very small. For completeness, we will amend the manuscript to provide these 6 pairs in the appendix.
> >
> >
> > > Q3: Similarity-Filtered Greedy Soup notation
> >
> > Thank you for drawing our attention to this omission. The symbol $\tau$ corresponds to the the task vector, and the subscripts i and j to the i-th and j-th checkpoint. We will update the manuscript to resolve this issue.
> >
> > > Q4: Clarifying “MonoSoup (Pairwise Models)” in Table 1
> >
> > In the pair-wise setting or in any multi-model setting, our method is applied to the model soups formulation (average of task vectors added to the pre-trained weights). Since our method operates on a single checkpoint, it is by design orthogonal to all multi-model methods. In this case, we opt for the simplest one.

---

> ### Author Response · Authors · 2025-11-25
>
> We thank Reviewer q1EA for their thoughtful review and for noting the relevance of the single-checkpoint problem as well as the appeal of the SVD-based interpretation. We provide below the clarifications and new experimental evidence that address the raised concerns.

---

> ### Comment · Reviewer_q1EA · 2025-11-27
>
> Thank you for the detailed clarification. Your response has resolved most of my concerns. I still have one remaining issue, however, and I believe that addressing it would significantly strengthen the paper.
>
> The alignment coefficient $\cos \alpha^{(\ell)} = \cos\big(W_0^{(\ell)}, \Delta W_{\text{Low}}^{(\ell)}\big)$ in the main paper (lines 266-268) plays a central role in the method and is interpreted as a measure of "pre-training knowledge preservation."
> However, this interpretation currently feels under-motivated: the paper defines a cosine between an absolute parameter state $W_0^{(\ell)}$ and a residual update $\Delta W_{\text{Low}}^{(\ell)}$, and directly equates this with preservation of pre-trained features, without showing a clear link to representation-level similarity or downstream behavior.
>
> This is conceptually unusual. In the literature, measures of "knowledge preservation" or "alignment" are typically instantiated as:
> - **state–state similarity** between pre-trained and fine-tuned parameters (e.g., shrinkage toward $W_0$ via L2-SP) [1], or
> - **update–update similarity** between task vectors when editing or merging models (e.g., cosine similarity between task vectors in task arithmetic) [2].
>
> I would therefore encourage the authors to provide quantitative evidence that higher alignment (state-update) indeed corresponds to better preservation of pre-training behavior. The exact form of such analysis is flexible, but examples include:
>
> - examining the correlation between layer-wise $\cos \alpha^{(\ell)}$ and CKA similarity to the pre-trained model; or
> - decomposing $\Delta W_{\text{Low}}^{(\ell)}$ into components parallel vs. orthogonal to $W_0^{(\ell)}$ and comparing their selective removal on ID accuracy and robustness.
>
> These are only examples. Any analysis that quantitatively links the proposed alignment to representation similarity or behavioral preservation would substantially clarify the conceptual role of $\cos \alpha^{(\ell)}$ and strengthen the main claim.
>
> `References`
>
> [1] Explicit Inductive Bias for Transfer Learning with Convolutional Networks, ICML 2018.
>
> [2] Editing Models with Task Arithmetic, ICLR 2023.

---

> > ### Author Response · Authors · 2025-11-27
> >
> > We thank the reviewer for pointing out that our original description of $\cos \alpha^{(\ell)}$ was conceptually unclear. In the revised manuscript we have taken two steps to clarify and justify its role.
> >
> > **1. Clarified definition of ($\cos \alpha^{(\ell)}$).**
> >
> > In Section 4 and Appendix C we now explicitly define $\cos \alpha^{(\ell)}$ as an *update-internal* quantity rather than a state–update cosine. Concretely, let $(W^{(\ell)} = \Delta W^{(\ell)})$ denote the fine-tuning update at layer (\ell), and let $W_{\text{high}}^{(\ell)}$ and $W_{\text{low}}^{(\ell)}$ be its high- and low-energy components obtained from the SVD in Eqs. (2)–(3). We define
> > $$
> > \cos^2 \alpha^{(\ell)} = \frac{\bigl|W_{\text{low}}^{(\ell)}\bigr|_F^2}{\bigl|W^{(\ell)}\bigr|_F^2},
> > $$
> > so that $\cos \alpha^{(\ell)}$ measures the fraction of the update’s squared Frobenius norm that lies in the low‑energy subspace (Eq. (7) and Theorem 1 in Appendix C).
> >
> > The mixing coefficient
> > $$
> > \lambda_{\text{low}}^{(\ell)} = \rho^{(\ell)} + \bigl(1 - \rho^{(\ell)}\bigr)\cos \alpha^{(\ell)}
> > $$
> > therefore increases exactly when low‑energy directions carry a substantial portion of the total update. This aligns the main text with the quantity actually analyzed in Appendix C; our earlier wording (“cosine between the pre-trained weights and the low‑energy subspace”) was imprecise and we will correct it.
> >
> > **2. Representation-level evidence via CKA.**
> >
> > To connect this scalar to “pre-training knowledge preservation,” we now explicitly point to a CKA analysis in **Appendix G (Figs. 8–9)**. For each transformer block (\ell) in both a weak (worst‑OOD) and a strong (best‑OOD) CLIP ViT‑B/32 checkpoint, we compute **linear CKA** between the pre-trained CLIP features and five variants: (i) Pre-trained, (ii) Fine-tuned, (iii) MonoSoup, (iv) High-only ($W_0 + \Delta W_{\text{high}}$), and (v) Low-only ($W_0 + \Delta W_{\text{low}}$), on both ImageNet (ID) and ImageNet‑A (OOD).
> >
> > This analysis reveals a consistent pattern:
> >
> > * The **High-only** model is nearly indistinguishable from the fine-tuned model in CKA, confirming that $\Delta W_{\text{high}}$ captures nearly all of the *representational drift away from pre-training*.
> > * The **Low-only** model remains closest to the pre-trained representation layer-wise, especially on OOD inputs, and exhibits **higher OOD robustness but reduced ID accuracy**.
> > * **MonoSoup** lies between these two extremes: it leaves early/mid layers almost unchanged relative to the fine-tuned model, but shifts deep layers noticeably back toward the pre-trained representation on ImageNet‑A, which aligns with its improved OOD accuracy while maintaining strong ID performance.
> >
> > Since $W_{\text{low}}^{(\ell)}$ is precisely the component whose influence is modulated by $\cos \alpha^{(\ell)}$ through $\lambda_{\text{low}}^{(\ell)}$, these CKA results provide representation-level evidence supporting our interpretation of $\cos \alpha^{(\ell)}$ as a **pre-training preservation** signal. In layers where low-energy updates are prominent (high $\cos \alpha^{(\ell)}$), increasing $\lambda_{\text{low}}^{(\ell)}$ places more weight on $W_{\text{low}}^{(\ell)}$, which our CKA analysis shows **moves the representation toward the pre-trained solution on OOD data**. In layers where low-energy mass is negligible (low $\cos \alpha^{(\ell)}$), $\lambda_{\text{low}}^{(\ell)}$ shrinks and MonoSoup essentially recovers the fine-tuned behavior.
> >
> > We will clarify this connection in the main text by (i) replacing the ambiguous “cosine with $W_0$” wording with the precise update-based definition above, and (ii) explicitly referencing the Appendix G CKA analysis as empirical evidence that the directions scaled by $\cos \alpha^{(\ell)}$ preserve pre-trained features and are responsible for OOD robustness.
> >
> >
> > We think the manuscript has been updated, and you may now view it. All analyses are indicated in blue.

---

> > > ### Comment · Reviewer_q1EA · 2025-11-28
> > >
> > > I appreciate the authors' detailed response and the clarification regarding the definition of $\cos \alpha$. However, it appears that the main text (sec.4) of the revised manuscript does not yet reflect these changes. I will review the updated manuscript once the revisions are properly incorporated, and I am willing to raise my score upon confirmation.

---

> > > > ### Author Response · Authors · 2025-11-28
> > > >
> > > > We thank the reviewer for their constructive comments. We have revised the main text in Section 4 to refine the wording, and these changes are highlighted in  $\textcolor{blue}{\text{blue}}$.
> > > >
> > > > Additionally, we have added a paragraph at the end of Appendix G, where the CKA results are discussed. This addition establishes a link between Section 4 and the CKA analysis, allowing the reader to gain a more precise intuition regarding the behavior of Equation 6 which, is presented in $\textcolor{violet}{\text{violet}}$.

---

### Official Review · Reviewer_YMQp · 2025-10-29

**Soundness:** 3
**Presentation:** 3
**Contribution:** 3
**Rating:** 6
**Confidence:** 3

**Summary:**

This paper presents MonoSoup, a light-weight alternative to model soup's traditional weight-space based ensembling techniques leveraging multiple model checkpoints. More specifically, by leveraging the SVD to estimate a high and low energy weight estimates, MonoSoup is able to re-estimate the final model. Empirical evaluations shows that MonoSoup achieves strong OOD Generalization performance when compared with traditional model souping techniques.

**Strengths:**

The reviewer notes the following strengths:
- The paper presents a clear context for MonoSoup with a defined motivation for the development of the underlying methodology.
- The proposed methodology (MonoSoup) is light-weight and readily applicable to real-world settings.
- MonoSoup showcases strong empirical performance across multiple models & tasks.
- The author also provide strong intuitive background for MonoSoup through analysis, linking performance improvements to alignment between fine-tuning updates.

**Weaknesses:**

The reviewer notes the following weaknesses:
- The reviewer’s primary concern is that, while the paper’s motivation is clearly stated, the argument that storing only a single best-performing checkpoint necessitates the development of MonoSoup is unconvincing. In particular, the reviewer finds it unlikely that, in practice, there would be meaningful constraints on retaining multiple checkpoints during model training.
- Additional evaluations on other modalities like audio would also provide even more compelling evidence for the applicability of MonoSoup.

**Questions:**

As noted in the weaknesses above, the reviewer encourages the authors to consider an alternative motivation for MonoSoup, rather than relying on the assumption that most models in practice need to be stored as single checkpoints.

---

> ### Author Response · Authors · 2025-11-25
>
> We thank the reviewer for their positive assessment and for highlighting the clarity, lightweight nature, and empirical strength of MonoSoup. We address the concerns below and provide refinements that strengthen the motivation and scope of the method. We address the reviewer’s concerns below.
>
> > W1 + Q: Clarifying MonoSoup motivation
>
> We appreciate the reviewer's perspective regarding storage constraints. While storage is indeed becoming cheaper, the main bottleneck for model ensembling is the computational cost to produce the initial model collection. As noted in the Model Stock paper, standard ensembling methods are often impractical because they require the training of dozens of fine-tuned models. Therefore, the overhead we aim to eliminate is primarily the GPU hours and energy required to train these models rather than required storage.
>
> Moreover, as we note in the paper, access to multiple suitable checkpoints is frequently unavailable; since public repositories often store a single best-performing version, which makes >2-model methods like ModelStock sometimes impractical.
>
> Finally, one of MonoSoup’s strengths is its versatility; it can be applied to any checkpoint, which can originate from a single fine-tuning run, or as the outcome of prior ensembling.
>
> > W2: Evaluating on additional modalities such as audio
>
> We appreciate the reviewer’s suggestion to evaluate across additional modalities such as audio. While the bulk of prior work on model-soup style approaches has focused on vision using the benchmark from the original Model Soups paper, our paper already takes a step beyond purely vision by including the LLM experiments on text with our QWEN experiments.
>
> We agree that applying MonoSoup to an audio benchmark would further strengthen generality — however, it is unclear how to define a similar benchmark: to our knowledge suitable large-scale, publicly accepted audio OOD benchmarks remain less standardized than for vision literature, and designing a meaningful fine-tune + OOD evaluation across audio remains unclear.

---

### Official Review · Reviewer_mj9q · 2025-10-31

**Soundness:** 2
**Presentation:** 2
**Contribution:** 2
**Rating:** 4
**Confidence:** 4

**Summary:**

The paper introduces MonoSoup, a post-hoc, data-free method that improves the trade-off between in-distribution (ID) accuracy and out-of-distribution (OOD) robustness using only a single fine-tuned model. Motivated by the geometric principles underlying model soups, the authors propose to analyze a single model’s fine-tuning update via Singular Value Decomposition (SVD), decomposing it into high-energy (task-specific) and low-energy (residual/robust) components. These components are adaptively reweighted using a combination of spectral decay and alignment with pretrained weights, yielding a single edited checkpoint that better balances specialization and generalization.

**Strengths:**

The paper tackles a practical and well-motivated challenge: retaining OOD robustness without storing or training multiple fine-tuned checkpoints. Its formulation is conceptually elegant, connecting the empirical success of model soups to the internal spectral geometry of a single model. The SVD-based decomposition provides an interpretable view of fine-tuning dynamics, distinguishing high-energy task adaptation from low-energy robustness-preserving directions. The adaptive weighting rule is simple, closed-form, and data-free, making the approach easy to implement as a lightweight post-processing step.

**Weaknesses:**

1. **Limited novelty beyond existing SVD-based merging.**

While the paper frames its contribution as extending model soups to a single-model setting, the actual main operation—SVD decomposition of the fine-tuning update followed by spectral weighting—closely parallels prior works (e.g., Task Singular Vectors, Model Merging with SVD). The paper’s novelty primarily lies in its interpretation rather than in a fundamentally new algorithmic principle.

2. **Heuristic coefficient design without theoretical grounding.**

The adaptive weighting rule $\lambda_{\text{low}} = \rho + (1-\rho) \cos(\alpha)$ is intuitively motivated but lacks theoretical justification for why this functional form optimally balances ID and OOD trade-offs. There is no analysis of stability, sensitivity, or convergence properties with respect to R, ρ, or α, leaving the approach partly empirical.

3. **Interpretational overrstatement of “soup”.**

The name MonoSoup implies model ensembling, yet the method operates as a single-model spectral reweighting rather than a true weight-space average. This conceptual mismatch may overstate its relation to model soups and obscure the fact that it performs deterministic, full-rank parameter editing rather than multi-model averaging.

**Questions:**

1. Beyond natural ImageNet shifts, could MonoSoup be evaluated on stronger distributional or adversarial robustness benchmarks to validate its claimed generalization benefits?

2. Since MonoSoup performs spectral reweighting rather than true model ensembling, what geometric evidence supports describing it as a “soup”? Would framing it as spectral anisotropic re-scaling be more accurate?

3. Considering prior fine-tuning mechanisms such as Spectral Adapter (Zhang & Pilanci, 2024) and other SVD-based spectral methods that already decompose pretrained weights to modulate singular directions, what specific conceptual or methodological advance distinguishes MonoSoup from this existing line of spectral fine-tuning approaches?

---

> ### Author Response · Authors · 2025-11-25
>
> We thank reviewer mj9q for their constructive feedback. We are glad that they deem our approach conceptually elegant and practical. We address their concerns below:
>
> > Limited novelty beyond existing SVD-based merging.
>
> We appreciate the reviewer’s careful comparison to prior SVD-based approaches. While SVD is indeed a broadly useful tool, the settings, motivations, and operations of these works **differ fundamentally** from ours. Prior SVD-based methods such as Task Singular Vectors and Model Merging with SVD operate in multi-task or multi-model regimes, where multiple checkpoints or tasks are available and the goal is to _extract shared structure_ across models. In contrast, MonoSoup addresses an orthogonal and previously unstudied problem: post-hoc editing from a single fine-tuned checkpoint, without data, training, or access to multiple models.
>
> We make this distinction explicit in the paper (**see Fig. 3**), where we show that the single-task, single-checkpoint setting exhibits _qualitatively different spectral behavior_ from multi-model merging. In particular, large-scale fine-tuning produces higher-rank updates with meaningful low-energy components—unlike the low-rank, noise-dominated residuals seen in small-scale multi-task settings. This difference is precisely why existing SVD truncation-based methods do not apply and why our high/low adaptive reweighting is required.
>
> MonoSoup is therefore not a restatement of prior SVD merging work, but a new formulation tailored to the single-checkpoint robustness problem, which has not been addressed by earlier SVD-based methods.
>
> >Heuristic coefficient design without theoretical grounding
>
> We thank the reviewer for their thoughtful question. Our goal is not to claim that $\lambda_{\text{low}}$ is globally optimal under all distributions, but rather that it is a *simple, principled, and empirically stable* rule derived from clear constraints. We will clarify this in the revision.
>
> First, we focus on the formula derivation. The coefficient $\lambda_{\text{low}} = \rho + (1 - \rho)\cos \alpha$ follows directly from four natural boundary conditions:
> - suppress low-energy components when spectrum is sharp and misaligned ($\rho \approx 0$, $\cos \alpha \approx 0$);
> - keep them when the spectrum is flat or strongly aligned ($\rho \approx 1$ or $\cos \alpha \approx 1$);
> - fall back to the spectral baseline when alignment is poor ($f(\rho, 0) = \rho$);
> - rely purely on alignment when low-energy mass is negligible ($f(0, \cos\alpha) = \cos\alpha$).
>
> If we furher assume that the interaction between $\rho$ and $\cos\alpha$ should be bilinear (the simplest smooth form), these four constraints *uniquely determine*
>
> $$
> f(\rho, \cos\alpha) = \rho + (1 - \rho)\cos\alpha .
> $$
>
> Thus the formula is not heuristic: it is the **minimal function** satisfying all desired behaviors.
>
> Regarding the reviewer's concerns about convergence and sensitivity, MonoSoup is a one-shot update, so convergence is not applicable. In terms of **sensitivity**, the rule is monotone and $1$-Lipschitz in both arguments:
>
> $$
> \frac{\partial f}{\partial \rho} = 1 - \cos\alpha \in [0, 1],
> \qquad
> \frac{\partial f}{\partial c} = 1 - \rho \in [0, 1].
> $$
>
> Hence small perturbations in $\rho$ or $\cos\alpha$ lead to proportionally small changes in
> $\lambda_{\text{low}}$.
>
> We will add this analysis to the revised manuscript. We hope this addresses the reviewer's concerns.
>
> > Interpretational overstatement of soup
>
> We agree with the reviewer that single-model spectral reweighting accurately conveys the proposed method. The fact that we operate on a single model is reflected on the **“Mono” prefix, while the “Soup”** part reflects that **our motivation comes from model soups** and related works. Specifically, not only the motivation of the proposed method comes from the original model soups paper but the methodology itself stems from **analyzing the geometrical and spectral properties of successful single-task model merging**. In our submitted abstract, we made an effort to differentiate from multi-model methods by highlighting that our method operates **“only a single checkpoint”** and that our method is **“practical and effective alternative to multi-checkpoint methods”**.
> We ask the reviewer to reconsider their score in light of this clarification.

---

> > ### Author Response · Authors · 2025-11-25
> > **Respond to the questions.**
> >
> > > Q1. Evaluating beyond natural ImageNet shifts.
> >
> > Our manuscript includes experiments beyond natural ImageNet shifts in section 5.2, where we evaluate the performance of MonoSoup on LLMs across a broad range of reasoning difficulties.  We propose this benchmark to go beyond the experiments solely on the image modality of previous works. Specifically, we consider GSM8K and SciQ, which overlap with the training mixture and are considered in-distribution tasks, as well as GSM_{Plus}, GSM8L_{Platinum}, and MMLU-Pro-Math, which assess advanced or adversarial reasoning skills not explicitly addressed during training, thereby serving as out-of-distribution evaluations.
> >
> >
> > > Q2: Spectral reweighting or soup?
> >
> > Building on our response to W2, we would like to reiterate that our method is motivated by model soups and an editing method rather than an ensembling method. The geometrical evidence that motivates our spectral reweighting does come from model soups in the proposed Similarity-Filtered Greedy Soup. We will update the manuscript to make the distinction more clear.
> >
> > > Q3: How is MonoSoup distinct from spectral adapters and other SVD-based spectral methods?
> > Fine-tuning mechanisms are fundamentally different from post-training methods such as MonoSoup. Our goal is to provide a post-hoc, data-free, single-checkpoint weight edit that reproduces the geometric effect of model soups inside a single model, without additional training, labels, or multiple checkpoints.
> >
> > Regarding comparisons to SVD-based approaches such as Task Singular Vectors and Model Merging with SVD, we note—as discussed in W1—that these methods operate in the multi-task, multi-checkpoint setting, where task vectors can be orthogonal or interfere. This setting is qualitatively different: low-energy components often correspond to noise across disparate tasks, and truncation behaves differently.
> >
> > In contrast, MonoSoup targets the single-task, single-checkpoint regime, where fine-tuning produces high-rank updates on large-scale datasets such as ImageNet, and where low-energy components preserve robustness-relevant signals rather than noise. As shown in Figure 3b, suppressing these components harms both ID and OOD which is a qualitative difference compared to truncation-based spectral methods and motivates our re-weighting formulation.
> >
> > MonoSoup therefore differs not only in the setting but also in intent: it performs anisotropic re-scaling of the fine-tuning update, rather than modulating pretrained weights or learning train-time adapters. To further underline this difference, we have added new CKA analyses in Appendix G, which show how MonoSoup selectively mitigates harmful representational drift in deep layers while preserving task-specific adaptation.
> >
> > Finally, following reviewer gZvH’s suggestion, we have also introduced an effective-rank, hyperparameter-free variant, which adapts the high/low split automatically and confirms that our approach remains robust without tuning. We hope this clarifies the conceptual and methodological differences with prior spectral methods and fully addresses the reviewer’s question.

---

> > > ### Comment · Reviewer_mj9q · 2025-11-26
> > >
> > > I appreciate the clarification regarding MonoSoup’s motivation and design.
> > >
> > > A remaining concern relates to scalability in the regime where the method is intended to provide the most practical benefit. All current experiments appear confined to small (ViT-S/32) and mid-sized (QWEN-0.6B) models (e.g., sub-billion or lightweight ViT small backbones), where the computational and memory demands of model soups are not yet prohibitive.
> > >
> > > The real bottleneck that motivates MonoSoup—storage and manipulation of multiple fine-tuned checkpoints—becomes dominant at the 2–7B scale and above, where SVD operations also become increasingly costly per layer.
> > >
> > > Without evidence at these scales, it remains unclear (i) whether the high/low spectral split remains stable, (ii) whether the reweighting continues to improve ID–OOD tradeoffs, and (iii) whether the overall runtime and memory footprint meaningfully outperform multi-checkpoint approaches.
> > >
> > > I therefore encourage adding at least one experiment on a 2–7B parameter model, ideally with both accuracy and cost metrics, to demonstrate that MonoSoup maintains its advantages in the intended large-model setting and does not introduce new bottlenecks of its own.

---

### Author Response · Authors · 2025-11-25

We thank the reviewers for their encouraging and constructive feedback. We are encouraged that the reviewers recognize the practicality and strong motivation behind our work (mj9q, YMQp, q1EA), describing the method as conceptually elegant (mj9q) and a creative reframing of model robustness (gzvH). We are also pleased that the reviewers found MonoSoup to be simple, lightweight, and efficient (mj9q, YMQp, Yqvj), supported by thorough analysis (gzvH, Yqvj) and strong empirical performance (YMQp).

 - We have provided individual responses to each reviewer. Additionally, we have incorporated new experimental results, which were prompted by the reviewers' suggestions, and briefly mention them below. All updates to the manuscript are marked in blue font. Detailed information regarding these additions can be found in the appendix of the revised manuscript.
Extended Performance Evaluation: Our method was tested on additional models, specifically ConvNeXt, to demonstrate broader applicability.


- **Ablation Study on $R$:** We conducted an ablation study to analyze the specific impact of the parameter $R$ within the ConvNeXt model architecture.


- **CKA Analysis (Addressing Reviewer qIEA):** We included a Centered Kernel Alignment (CKA) study, complete with heatmaps, to provide empirical support for our claims regarding $W_{\text{low}}$ and $W_{\text{high}}$.


- **Novel Variant Proposal (Addressing Reviewer gzvH):** We introduced and evaluated a new variant of our method that eliminates the need for a variance threshold. This variant was benchmarked on CLIP, and ConvNeXt models.

---

### Meta-Review · Area_Chair_grRc · 2025-12-16

**Summary:**

The paper proposes a post hoc editing approach for fine-tuned model merging that aims to improve both in-distribution (ID) and out-of-distribution (OOD) performance by reweighting high- and low-energy components obtained via singular value decomposition (SVD). The paper is generally well motivated and clearly presented, and the proposed method offers an efficient post-processing strategy applicable to existing fine-tuned checkpoints.

However, reviewers raised several concerns. In particular, the SVD-based formulation is perceived as having limited novelty relative to existing methods, the design of the weighting strategy lacks theoretical justification, the reported gains over prior work are modest, and the experimental analysis is limited in scope. In the rebuttal, the authors clarified the relationship to prior SVD-based approaches and provided additional experimental results. Nonetheless, the AC finds that the explanation for the weighting parameter $\lambda_\text{low}$​ remains largely empirical, with no supporting theoretical analysis. The AC also concurs with the reviewers that the improvements over existing methods are relatively modest.

Consequently, the AC recommends **rejection** and encourages the authors to strengthen the theoretical motivation and deepen the analysis in future revisions.

**Reviewer Concerns:**

During the rebuttal, the authors addressed several major concerns.

1. Regarding novelty beyond existing SVD-based merging methods, the authors clarified that the proposed approach is tailored to single-checkpoint robustness, a setting in which prior SVD truncation–based methods are not directly applicable.

2. The authors provided clarifications on the motivation, experimental setup, and method description.

3. Additional experimental analyses were included, covering representation alignment, effective-rank–informed design choices, and diverse architectures and settings.

However, two major concerns remain insufficiently addressed.

1. The weighting design for $\lambda_\text{low}$​ lacks theoretical support. Although the authors argue that the design satisfies four natural boundary conditions, they do not clearly justify why these conditions are necessary for alignment. Moreover, the claim that
$W_\text{low}$ is critical for OOD robustness is based primarily on empirical observations, without a supporting theoretical foundation.

2. While modest gains may be expected from a single-step post hoc editing procedure, the paper does not explore alternative or more expressive settings (e.g., multi-step editing during fine-tuning process) that could potentially yield stronger improvements.

**Reviewer Scores:**

Below are the updated scores for each reviewer based on the discussion. mj9q: 4; YMQp: 6; q1EA: 4$\rightarrow$6; gzvH: 6$\rightarrow$8; Yqvj: 6$\rightarrow$8.

---

### Decision · Program_Chairs · 2026-01-26

Reject